# DUAL-FORECASTER: A MULTIMODAL TIME SERIES MODEL INTEGRATING DESCRIPTIVE AND PREDICTIVE TEXTS

## ABSTRACT

Time series forecasting plays a vital role for decision-making across a wide range of real-world domains, which has been extensively studied. Most existing single-modal models rely solely on numerical series, which suffer from the limitations imposed by insufficient information. Recent studies have revealed that multi-modal models can address the core issue by integrating textual information. However, these models focus on either historical or future textual information, over-looking the unique contributions each plays in time series forecasting. Besides, these models fail to grasp the intricate relationships between textual and time series data, constrained by their moderate capacity for multimodal comprehension. To tackle these challenges, we propose Dual-Forecaster, a pioneering multimodal time series model that combines both descriptively historical textual information and predictive textual insights, leveraging advanced multimodal comprehension capability. We begin by developing the historical text-time series contrastive loss to align the descriptively historical textual data and corresponding time series data, followed by encoding multimodal text-time series representations between them through the history-oriented modality interaction module, and then combining predictive textual data through the future-oriented modality interaction module to ensure textual insights-following forecasting. Our comprehensive evaluations on synthetic dataset and captioned-public datasets demonstrate that Dual-Forecaster is a distinctly effective multimodal time series model that outperforms or is comparable to other state-of-the-art models, highlighting the superiority of integrating textual information for time series forecasting. This work opens new avenues in the integration of textual information with numerical time series data for multi-modal time series analysis.

## 1 INTRODUCTION

With the massive accumulation of time series data in such various domains as retail (Leonard, 2001), electricity (Liu et al., 2023a), traffic (Shao et al., 2022), finance (Li et al., 2022), and health-care (Kaushik et al., 2020), time series forecasting has become a key part of decision-making. To date, while extensive research has been dedicated to time series forecasting, resulting in a multi-tude of proposed methodologies (Hyndman et al., 2008; Nie et al., 2022; Liu et al., 2023b; Garza & Mergenthaler-Canseco, 2023; Xue & Salim, 2023; Zhou et al., 2023), they are predominantly confined to single-modal models that rely exclusively on numerical time series data. Constrained by the scarcity information within time series data, these models have hit a roadblock in improving the forecasting performance due to overfitting on the training data.

To improve the model's forecasting performance, it is crucial to introduce supplementary informa-tion that is not present in time series data. For instance, when forecasting future product sales, combining numerical historical sales data with external factors—such as product iteration plans, strategic sales initiatives, and unforeseeable occurrences like pandemics—enables us to give a sales forecast that aligns more closely with business expectations. This supplementary information usu-ally manifests as unstructured text, rich in semantic information that reflects temporal causality and system dynamics. In contrast to quantifiable information like holidays can be readily embedded as covariates through feature engineering techniques, this supplementary text is challenging to distill

into numerical format, posing a barrier to its utilization in bolstering the reliability of time series forecasts.

Recently, there has been a surge in proposals for multimodal time series models that incorporate text as an additional input modality (Xu et al., 2024; Liu et al., 2024b;a). This methodology effectively surpasses the constraints inherent in traditional time series forecasting approaches, substantially augmenting the accuracy and efficacy of the models. However, these models often focus exclusively on historical or future textual information, thereby underestimating the distinct roles each type of information plays in time series forecasting. Moreover, they struggle to capture the complex connections between textual and time series data, due to their constrained multimodal comprehension capabilities. Therefore, it is necessary to integrate both historical and future textual data and enhance the model's capacity for multimodal understanding.

To tackle the aforementioned challenges, we introduce Dual-Forecaster, a cutting-edge time series forecasting model. It is designed around a sophisticated framework that adeptly aligns historical and future textual data with time series data, capitalizing on its robust multimodal comprehension capability. It should be noted that the word 'Dual' in Dual-Forecaster has two different levels of meaning. On the one hand, it represents that Dual-Forecaster is a multimodal time series model capable of handling both textual and time series data concurrently, and on the other hand, it denotes the model's capacity to simultaneously process descriptively historical textual data and predictive textual data, integrating these with time series data within a unified high-dimension embedding space to better guide the forecasting. Specifically, Dual-Forecaster consists of the textual branch and the temporal branch. The textual branch is responsible for understanding textual data and extracting the semantic information contained within it, while the temporal branch processes time series information. To improve the model's multimodal comprehension capability, we propose three well-designed cross-modality alignment techniques: (1) **Historical Text-Time Series Contrastive Loss** aligns the descriptively historical textual data and corresponding time series data, augmenting the model's capacity to learn inter-variable relationships; (2) **History-oriented Modality Interaction Module** integrates input historical textual and time series data through a cross-attention mechanism, ensuring effective alignment of distributions between historical textual and time series data; (3) **Future-oriented Modality Interaction Module** incorporates the predictive textual insights into the aligned time series embeddings generated by history-oriented modality interaction module, ensuring textual insights-following forecasting for obtaining more reasonable forecasts.

To prove the effectiveness of our model, we conduct extensive experiments on synthetic dataset and captioned-public datasets. Experimental results demonstrate that Dual-Forecaster achieves competitive or superior performance when compared to other state-of-the-art models on all datasets. Moreover, ablation studies emphasize that the performance enhancement is attributed to the supplementary information provided by both descriptively historical textual data and predictive textual data.

Our main contributions in this work are threefold:

(1) We craft a sophisticated framework for the integration of textual and time series data underpinned by advanced multimodal comprehension. This framework is engineered to generate time series embeddings that are enriched with enhanced semantic insights, which in turn, empowers Dual-Forecaster a more robust forecasting capability.

(2) We propose Dual-Forecaster, a novel time series forecasting model that excels in integrating descriptively historical text with time series data, thereby enhancing the model's capacity to discern inter-variable relationships. Additionally, it utilizes predictive textual insights to direct the forecasting process, subsequently bolstering the reliability of forecasts.

(3) Extensive experiments on synthetic dataset and captioned-public datasets demonstrate that Dual-Forecaster achieves state-of-the-art performance on time series forecasting task, with favorable generalization ability.

## 2 RELATED WORK

**Time series forecasting.** Time series forecasting models can be roughly categorized into statistical models and deep learning models. Statistical models such as ETS, ARIMA (Hyndman et al., 2008)

can be fitted to a single time series and used to make predictions of future observations. Deep learning models, ranging from the classical LSTM (Hochreiter, 1997), TCN (Bai et al., 2018), to recently popular transformer-based models (Nie et al., 2022; Zhou et al., 2022; Zhang & Yan, 2023; Liu et al., 2023b), are developed for capturing nonlinear, long-term temporal dependencies. Even though excellent performance has been achieved on specific tasks, these models lack generalizability to diverse time series data.

To overcome the challenge, the development of pre-trained time series foundation models has emerged as a burgeoning area of research. In the past two years, several time series foundation models have been introduced (Garza & Mergenthaler-Canseco, 2023; Rasul et al., 2023; Das et al., 2023; Ansari et al., 2024; Woo et al., 2024). All of them are pre-trained transformer-based models trained on a large corpus of time series data with time-series-specific designs in terms of time features, time series tokenizers, distribution heads, and data augmentation, among others. These pre-trained time series foundation models can adapt to new datasets and tasks without extensive from-scratch retraining, demonstrating superior zero-shot forecasting capability. Furthermore, benefiting from the impressive capabilities of pattern recognition, reasoning and generalization of Large Language Models (LLMs), recent studies have further explored tailoring LLMs for time series data through techniques such as fine-tuning (Xue & Salim, 2023; Gruver et al., 2024; Zhou et al., 2023) and model reprogramming (Cao et al., 2023; Jin et al., 2023; Pan et al., 2024; Sun et al., 2023). However, existing time series forecasting models have encountered a plateau in performance due to limited information contained in time series data. There is an evident need for additional data beyond the scope of time series to further refine forecasts.

**Text-guided time series forecasting**. Some works have attempted to address the prevalent issue of information insufficiency in the manner of text-guided time series forecasting, which includes text as an auxiliary input modality. A line of work investigate how to use some declarative prompts (*e.g.*, domain expert knowledge and task instructions) enriching the input time series to guide LLM reasoning (Jin et al., 2023; Liu et al., 2024d;c). These approaches align text and time series into the language space based on the global features learned by pre-trained LLMs to maximize the inference capability of the LLMs. However, they ignore the importance of local temporal features on time series forecasting.

An alternative text-guided time series forecasting approach is to process textual and time series data separately by using different models, and then merge the information of two modalities through a modality interaction module to yield enriched time series representations for time series forecasting (Liu et al., 2024a). Our method belongs to this category, however, there is limited relevant research on time series. Xu et al. (2024) proposes TGForecaster that employs a PatchTST encoder to process time series data and utilizes off-the-shelf, pre-trained text models to pre-embedding news content and channel descriptions into vector sequences across the time dimensions, thus allowing for efficient modality fusion in the embedding space. Liu et al. (2024b) develops MM-TFSlib, which provides a convenient multimodal integration framework. It can independently model numerical and textual series using different time series forecasting models and LLMs, and then combine these outputs using a learnable linear weighting mechanism to produce the final predictions. Distinct from these methods, we recognize the unique contributions of historical and future textual information in time series forecasting, as well as the significance of multimodal comprehension capability in learning the intricate links between textual and time series data. We introduce a groundbreaking multimodal time series model that integrates both descriptively historical textual information and predictive textual insights based on advanced multimodal comprehension capability.

## 3 METHODOLOGY

### 3.1 PROBLEM FORMULATION

Given a dataset of $N$ numerical time series and their corresponding textual series, $\mathcal{D} = \{(X_{t-L:t}^{(i)}, X_{t:t+h}^{(i)}, S_{t-L:t}^{(i)}, S_{t:t+h}^{(i)})\}_{i=1}^{N}$, where $X_{t-L:t}^{(i)}$ is the input variable of the numerical time series, $L$ is the specified *look back window* length, and $X_{t:t+h}^{(i)}$ is the ground truth of *horizon window* length $h$. $S_{t-L:t}^{(i)}$ is the overall description of $X_{t-L:t}^{(i)}$, which can be used to augment the model's capacity to learn the relationships between different time series by combining detailed descrip-

tive information about the time series. $\mathbf{S}_{t:t+h}^{(i)}$ is the overall description of $\mathbf{X}_{t:t+h}^{(i)}$, which can provide additional predictive insights to assist the model in perceiving and dynamically adapting to event-driven time series distribution drift. The goal is to maximize the log-likelihood of the predicted distribution $p\left(\mathbf{X}_{t:t+h}|\hat{\boldsymbol{\phi}}\right)$ obtained from the distribution parameters $\hat{\boldsymbol{\phi}}$ learned by the model $\boldsymbol{f_\theta} : (\mathbf{X}_{t-L:t}, \mathbf{S}_{t-L:t}, \mathbf{S}_{t:t+h}) \to \hat{\boldsymbol{\phi}}$ based on historical time series data and its corresponding descriptive textual information and predictive textual insights:

$$
\max_{\boldsymbol{\theta}} \mathbb{E}_{(\mathbf{X},\mathbf{S})\sim p(\mathcal{D})} \log p\left(\mathbf{X}_{t:t+h}|\hat{\boldsymbol{\phi}}\right)
$$
$$
s.t. \hat{\boldsymbol{\phi}} = \boldsymbol{f_\theta} : (\mathbf{X}_{t-L:t}, \mathbf{S}_{t-L:t}, \mathbf{S}_{t:t+h})
$$
(1)

where $p(\mathcal{D})$ is the data distribution used for sampling numerical time series and their corresponding textual series.

## 3.2 ARCHITECTURE

Illustrated in Figure 1, our proposed Dual-Forecaster consists of two branches: the textual branch and the temporal branch. The textual branch comprises a pre-trained *RoBERTa* model (Liu, 2019) and an attentional pooler. The frozen pre-trained *RoBERTa* model is responsible for tokenization, encoding, and embedding of text. The attentional pooler is adopted to customize text representations produced by *RoBERTa* model to be used for different types of training objectives (Lee et al., 2019; Yu et al., 2022). The temporal branch is composed of a unimodal time series encoder for processing time series information and modality interaction modules with PaLM (Chowdhery et al., 2023) as the backbone. The unimodal time series encoder is used for patching and embedding of time series. It is noteworthy that the [CLS] as a global representation of time series is introduced into the embedded representation vector. The modality interaction modules are utilized to discern the relationships between textual and time series data, thereby enhancing the capacity for multimodal understanding. In concrete, the textual branch respectively takes the historical text $\mathbf{S}_{t-L:t}$ and the future text $\mathbf{S}_{t:t+h}$ as inputs to obtain their corresponding embeddings $\widetilde{\mathbf{S}}_{q(t-L:t)}$, $\widetilde{\mathbf{S}}_{CLS(t-L:t)}$, and $\widetilde{\mathbf{S}}_{q(t:t+h)}$, $\widetilde{\mathbf{S}}_{CLS(t:t+h)}$. The temporal branch works with the historical time series $\mathbf{X}_{t-L:t}$ to obtains its corresponding embedding $\widetilde{\mathbf{X}}_{P(t-L:t)}$, $\widetilde{\mathbf{X}}_{CLS(t-L:t)}$. To improve the model's multimodal comprehension capability, we implement three distinct cross-modality alignment techniques: historical text-time series contrastive loss, historical-oriented modality interaction module, and future-oriented modality interaction module. In the following section, we will elaborate on the descriptions of these techniques.

### 3.2.1 HISTORICAL TEXT-TIME SERIES CONTRASTIVE LOSS

Previous multimodal time series models, whether they integrate historical texts like descriptions of input time series (Liu et al., 2024a) or future texts such as news and channel descriptions (Xu et al., 2024), typically utilize separate text and temporal branches to process their respective modality data. Subsequently, they employ a cross-attention-based modality interaction module to facilitate the integration of these distinct modality data. Given that the text features and time series embeddings reside in their own high-dimensional spaces, it is challenging for these models to effectively learn and model their interactions. While model reprogramming techniques like Time-LLM (Jin et al., 2023) align time series representations into the language space, thus unleashing the potential of LLM as a predictor, these approaches often overlook the critical role of local temporal features.

Inspired by the VLP framework in CV (Li et al., 2021), in this work, we attempt to align text features and time series embeddings into the unified high-dimensional space before fusing in the modality interaction modules. Therefore, we develop a historical text-time series contrastive loss to deal with this problem. Specifically, for each input time series $\mathbf{X}_{t-L:t}^{(i)} \in \mathbb{R}^{1\times L}$, it is first normalized to have zero mean and unit standard deviation in mitigating the time series distribution shift. Then, we divide it into $P$ consecutive non-overlapping patches with length $L_p$. Given these patches $\mathbf{X}_{P(t-L:t)}^{(i)} \in \mathbb{R}^{P\times L_p}$, we adopt a simple linear layer to embed them as $\hat{\mathbf{X}}_{P(t-L:t)}^{(i)} \in \mathbb{R}^{P\times d_m}$, where $d_m$ is the dimensions of time series features. On this basis, we introduce the time series CLS token $\hat{\mathbf{X}}_{CLS(t-L:t)}^{(i)} \in \mathbb{R}^{1\times d_m}$. Let $\hat{\mathbf{X}}_{t-L:t}^{(i)} = \left[\hat{\mathbf{X}}_{P(t-L:t)}^{(i)} \; \hat{\mathbf{X}}_{CLS(t-L:t)}^{(i))}\right] \in \mathbb{R}^{(P+1)\times d_m}$. We use the $n_{uni}$

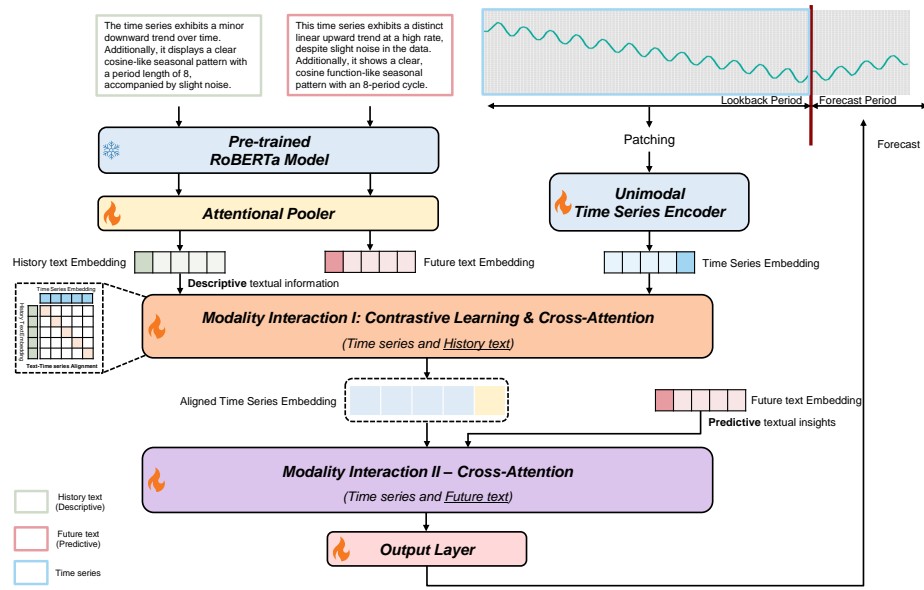

Figure 1: Overall architecture of Dual-Forecaster. Top left is the text branch with text input, and top right is the temporal branch with time series input. Based on the obtained text features and time series embeddings, to improve the model's multimodal comprehension capability, we employ three cross-modality alignment techniques: historical text-time series contrastive loss and history-oriented modality interaction module (***Modality Interaction I***), as well as future-oriented modality interaction module (***Modality Interaction II***). The outputs of time series embeddings from ***Modality Interaction II*** are projected to generate the final forecasts.

layers of PaLM containing *Multi-Head Self-Attention* (*MHSA*) layers to process time series, and finally take the outputs of the $n_{uni}^{th}$ layer as the embeddings $\widetilde{\boldsymbol{X}}_{t-L:t}^{(i)} \in \mathbb{R}^{(P+1) \times d_m}$:

$$\widetilde{\boldsymbol{X}}_{t-L:t}^{(i)} = \left( MHSA \left( \hat{\boldsymbol{X}}_{t-L:t}^{(i)} \right) + \hat{\boldsymbol{X}}_{t-L:t}^{(i)} \right)_{n_{uni}^{th}} = \left[ \widetilde{\boldsymbol{X}}_{P(t-L:t)}^{(i)} \ \widetilde{\boldsymbol{X}}_{CLS(t-L:t)}^{(i)} \right] \tag{2}$$

For each historical text $\boldsymbol{S}_{t-L:t}^{(i)}$, we use the pre-trained *RoBERTa* model for tokenization, encoding, and embedding to obtain $\hat{\boldsymbol{S}}_{G(t-L:t)}^{(i)} \in \mathbb{R}^{G \times d}$, where $G$ represents the number of tokens encoded in the historical text and $d$ is the dimensions of text features. On this basis, we introduce learnable text query $\hat{\boldsymbol{Q}}_{q}^{(i)} \in \mathbb{R}^{q \times d_m}$ and text CLS token $\hat{\boldsymbol{Q}}_{CLS}^{(i)} \in \mathbb{R}^{1 \times d_m}$. Let $\hat{\boldsymbol{Q}}^{(i)} = \left[ \hat{\boldsymbol{Q}}_{q}^{(i)} \ \hat{\boldsymbol{Q}}_{CLS}^{(i)} \right] \in \mathbb{R}^{(q+1) \times d_m}$. We use a *Multi-Head Cross-Attention* (*MHCA*) layer with $\hat{\boldsymbol{Q}}^{(i)}$ as query and $\hat{\boldsymbol{S}}_{G(t-L:t)}^{(i)}$ as key and value to obtain the embedding $\widetilde{\boldsymbol{S}}_{t-L:t}^{(i)} \in \mathbb{R}^{(q+1) \times d_m}$:

$$\widetilde{\boldsymbol{S}}_{t-L:t}^{(i)} = MHCA \left( \hat{\boldsymbol{Q}}^{(i)}, \hat{\boldsymbol{S}}_{G(t-L:t)}^{(i)} \right) = \left[ \widetilde{\boldsymbol{S}}_{q(t-L:t)}^{(i)} \ \widetilde{\boldsymbol{S}}_{CLS(t-L:t)}^{(i)} \right] \tag{3}$$

Similarly, for each future text $\boldsymbol{S}_{t:t+h}^{(i)}$, we can obtain the embedding $\widetilde{\boldsymbol{S}}_{t:t+h}^{(i)} \in \mathbb{R}^{(q+1) \times d_m}$ in the manner described above:

$$\widetilde{\boldsymbol{S}}_{t:t+h}^{(i)} = MHCA \left( \hat{\boldsymbol{Q}}^{(i)}, \hat{\boldsymbol{S}}_{G(t:t+h)}^{(i)} \right) = \left[ \widetilde{\boldsymbol{S}}_{q(t:t+h)}^{(i)} \ \widetilde{\boldsymbol{S}}_{CLS(t:t+h)}^{(i)} \right] \tag{4}$$

Given the outputs of $\widetilde{\boldsymbol{S}}_{CLS(t-L:t)}^{(i)}$ and $\widetilde{\boldsymbol{X}}_{CLS(t-L:t)}^{(i)}$ from the text branch and the unimodal time series encoder of the temporal branch, respectively, the historical text-time series contrastive loss is

defined as:

$$\text{sim}_i = \widetilde{\boldsymbol{X}}_{CLS(t-L:t)}^{(i)} \bigodot \widetilde{\boldsymbol{S}}_{CLS(t-L:t)}^{(i)}$$

$$\mathcal{L}_{contrastive} = -\frac{1}{B} \left( \sum_i^B \log \frac{exp\left(\text{sim}_i^T y_i/\tau\right)}{\sum_{j=1}^B exp\left(\text{sim}_j^T y_j\right)} + \sum_i^B \log \frac{exp\left(y_i^T \text{sim}_i/\tau\right)}{\sum_{j=1}^B exp\left(y_j^T \text{sim}_j\right)} \right) \tag{5}$$

where $B$ is the batch size, $y_i \in \mathbb{R}^{B \times B}$ is the one-hot label matrix from ground truth text-time series pair label, and $\tau$ is the temperature to scale the logits.

### 3.2.2 HISTORY-ORIENTED MODALITY INTERACTION MODULE

To ensure effective alignment of distributions between historical textual and time series data, we use the $n_{mul}$ layers of PaLM, including a *MHSA* operation and a *MHCA* operation in each layer, as the history-oriented modality interaction module to obtain the aligned time series embedding that integrates historical textual information. Formally, given $\widetilde{\boldsymbol{S}}_{q(t-L:t)}^{(i)}$ and $\widetilde{\boldsymbol{X}}_{P(t-L:t)}^{(i)}$ produced by the textual branch and the unimodal time series encoder of the temporal branch, at each layer of the history-oriented modality interaction module, we sequentially process and aggregate the textual and temporal information based on the *MHSA* and *MHCA* mechanism, and finally take the outputs of the $n_{mul}^{th}$ layer as the aligned time series embeddings $\bar{\boldsymbol{X}}_{align}^{(i)} \in \mathbb{R}^{P \times d_m}$:

$$\bar{\boldsymbol{X}}_{align}^{(i)} = \left( MHCA \left( MHSA \left( \widetilde{\boldsymbol{X}}_{P(t-L:t)}^{(i)} \right) + \widetilde{\boldsymbol{X}}_{P(t-L:t)}^{(i)}, \widetilde{\boldsymbol{S}}_{q(t-L:t)}^{(i)} \right) + \widetilde{\boldsymbol{X}}_{P(t-L:t)}^{(i)} \right)_{n_{mul}^{th}} \tag{6}$$

### 3.2.3 FUTURE-ORIENTED MODALITY INTERACTION MODULE

To combine the predictive future text with the aligned time series embeddings produced by the history-oriented modality interaction module, we employ a *MHCA* layer as the future-oriented modality interaction module. This approach allows us to derive the time series embeddings that further integrate future textual information. In concrete, given the outputs of $\widetilde{\boldsymbol{S}}_{q(t:t+h)}^{(i)}$ and $\bar{\boldsymbol{X}}_{align}^{(i)}$ from the textual branch and history-oriented modality interaction module, respectively, we have the *MHCA* operation with $\bar{\boldsymbol{X}}_{align}^{(i)}$ as query and $\widetilde{\boldsymbol{S}}_{q(t:t+h)}^{(i)}$ as key and value to obtain the final time series embeddings $\bar{\boldsymbol{X}}_{final}^{(i)} \in \mathbb{R}^{P \times d_m}$ that integrate both descriptively historical textual information and predictive textual insights:

$$\bar{\boldsymbol{X}}_{final}^{(i)} = MHCA \left( \bar{\boldsymbol{X}}_{align}^{(i)}, \widetilde{\boldsymbol{S}}_{q(t:t+h)}^{(i)} \right) + \bar{\boldsymbol{X}}_{align}^{(i)} \tag{7}$$

### 3.2.4 OUTPUT PROJECTION

To maintain homogeneity with $\mathcal{L}_{contrastive}$, we use negative log-likelihood loss as the forecast loss, which constrains the model's predicted distribution to closely align with the actual distribution. Specifically, given $\bar{\boldsymbol{X}}_{final}^{(i)}$, we linearly project the last token embedding $\bar{\boldsymbol{X}}_{final[-1]}^{(i)} \in \mathbb{R}^{1 \times d_m}$ to obtain the distribution parameters of the Student's T-distribution prediction head. The forecast loss used is defined as:

$$\mathcal{L}_{forecast} = -\frac{1}{B} \sum_i^B \log p \left( \boldsymbol{X}_{t:t+h}^{(i)} | \hat{\phi} \left( \bar{\boldsymbol{X}}_{final[-1]}^{(i)} \right) \right) \tag{8}$$

The overall loss during training is the summation of the forecast loss $\mathcal{L}_{forecast}$ and the contrastive loss $\mathcal{L}_{contrastive}$ as follows:

$$\mathcal{L} = \mathcal{L}_{forecast} + \mathcal{L}_{contrastive} \tag{9}$$

## 4 MAIN RESULTS

To demonstrate the effectiveness of the proposed Dual-Forecaster, we firstly design five multimodal time series benchmark datasets across two categories: synthetic dataset and captioned-public dataset,

and then conduct extensive experiments on them. More details on the construction of the multi-modal time series benchmark datasets are in Appendix B.2. We compare Dual-Forecaster against a collection of representative methods from the recent time series forecasting landscape, our approach displays competitive or stronger results in multiple benchmarks and zero-shot setting.

**Baseline Models.** We carefully select 6 forecasting methods as our baselines with the following categories: (1) *Single-modal models*: DLinear (Zeng et al., 2023), FITS (Xu et al., 2023), PatchTST (Nie et al., 2022), iTransformer (Liu et al., 2023b); (2) *Multimodal models*: MM-TSFlib (Liu et al., 2024b) with GPT-2 (Radford et al., 2019) as LLM backbone and iTransformer as time series fore-casting backbone (hereinafter referred to as MM-TSFlib), and Time-LLM (Jin et al., 2023). We contrast Dual-Forecaster with the *single-modal models* to illustrate how textual insights can enhance forecasting performance. Comparisons with the *multimodal models* highlight that Dual-Forecaster possesses enhanced multimodal comprehension capability, thereby further elevating the model's forecasting performance. More details are in Appendix C.

## 4.1 EVALUATION ON SYNTHETIC DATASET

**Setups.** The synthetic dataset is adopted to assess the model's capacity to utilize textual infor-mation for time series forecasting while effectively mitigating distribution drift. It is composed of simulated time series data containing different proportions of trend, seasonality, noise components, and switch states. Detailed descriptions of this dataset are provided in Appendix B.2.1. For a fair comparison, the input time series *look back window* (LBW) length $L$ is set as 200, and the prediction horizon $h$ is set as 30. Consistent with prior works, we choose the Mean Square Error (MSE) and Mean Absolute Error (MAE) as the default evaluation metrics.

**Results.** Table 1 presents the performance comparison of various models on synthetic dataset. Our model consistently outperforms all baseline models. The comparison with MM-TFSlib(Liu et al., 2024b) is particularly noteworthy. MM-TFSlib is a very recent work that provides a convenient multimodal integration framework for freely integrating open-source language models and various time series forecasting models. We note that our approach reduces MSE/MAE by **14.35%/13.21%** compared to MM-TSFlib. When compared with SOTA Transformer-based model PatchTST (Nie et al., 2022), we observe a reduction of **14.38%/12.81%** in MSE/MAE.

Table 1: Forecasting result on synthetic dataset. The best and second best results are in **bold** and underlined.

| Methods | | Dual-Forecaster | | DLinear | | FITS | | PatchTST | | iTransformer | | MM-TSFlib | | Time-LLM | |
|---|---|---|---|---|---|---|---|---|---|---|---|---|---|---|---|
| Metric | | MSE | MAE | MSE | MAE | MSE | MAE | MSE | MAE | MSE | MAE | MSE | MAE | MSE | MAE |
| synthetic dataset | 30 | **0.5150** | **0.4703** | 1.2190 | 0.8139 | 2.7585 | 1.3254 | 0.6015 | 0.5394 | 0.6190 | 0.5529 | 0.6013 | 0.5419 | 0.8907 | 0.6976 |

**Case Study.** Figure 2 shows an example of a simulated time series in synthetic dataset, which exhibits a trend transition from downward to upward within the forecasting horizon, simultaneously keeping the seasonality unchanged. In this visualization, both PatchTST and MM-TSFlib maintain the state observed in the *look back window*, indicating an inability to adapt to state transitions. No-tably, in the absence of textual information input, Dual-Forecaster also tends to provide conservative forecasts akin to these two methods. In other words, the generated forecasts are extensions of time series patterns observed in the historical time series data. Conversely, with the integration of textual insights, Dual-Forecaster can adaptively perceive potential future state transitions, thereby deliver-ing more reasonable forecasts. This underscores the substantial benefits of incorporating textual data for time series forecasting.

## 4.2 EVALUATION ON CAPTIONED-PUBLIC DATASETS

**Setups.** The captioned-public datasets are utilized to evaluate the model's capability of better per-forming time series forecasting by combining textual information to eliminate uncertainty in com-plex time series scenarios. They consist of the captioned version of ETTm1, ETTm2, ETTh1, and ETTh2, which have been extensively adopted for benchmarking various time series forecasting mod-els. Additionally, we include two challenging datasets of exchange-rate and stock indices, which better represent real-world scenarios. More details about the captioning process of these public

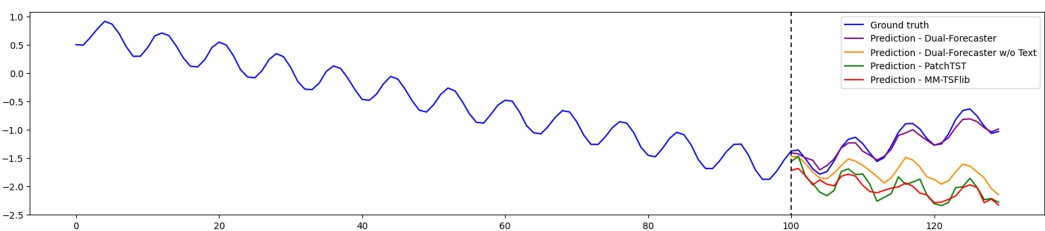

Figure 2: Visualization of an example from synthetic dataset under the input-100-predict-30 settings.

datasets are provided in the Appendix B.2.2. It should be noted that due to resource constraints, we construct 6 relatively small datasets on the basis of these datasets by setting the value of stride and conduct experiments on them. For ETTm1 and ETTm2 datasets, stride is set to 16, while for ETTh1 and ETTh2 datasets, stride is fixed as 4. For exchange-rate dataset, stride is set to 12, while for stock dataset, stride is fixed as 32. In this case, the input time series *look back window* length $L$ is set to 336, and the prediction *horizon $h$* is fixed as 96. It should be noted that for the stock dataset, the prediction *horizon $h$* is fixed as 21 (approximately one month of trading day).

**Results.** As demonstrated in Table 10, Dual-Forecaster consistently surpasses all baselines by a large margin, over **15.1%/12.3%** w.r.t. the second-best in MSE/MAE reduction.

Table 2: Forecasting result on captioned-public datasets.The best result is highlighted in **bold** and the second best is highlighted in underlined.

| Methods | | Dual-Forecaster | | DLinear | | FITS | | PatchTST | | iTransformer | | MM-TSFlib | | Time-LLM | |
|---|---|---|---|---|---|---|---|---|---|---|---|---|---|---|---|---|
| Metric | | MSE | MAE | MSE | MAE | MSE | MAE | MSE | MAE | MSE | MAE | MSE | MAE | MSE | MAE |
| ETTm1 | 96 | **1.2126** | **0.7686** | 1.5601 | 0.9198 | 2.2858 | 1.1810 | 1.4544 | 0.8619 | 1.3393 | 0.8299 | 1.3620 | 0.8426 | 1.4457 | 0.8730 |
| ETTm2 | 96 | **0.8469** | **0.5756** | 1.1663 | 0.7332 | 1.7418 | 0.9709 | 0.9419 | 0.6280 | 1.0210 | 0.6557 | 1.0325 | 0.6691 | 1.1199 | 0.7054 |
| ETTh1 | 96 | **1.4190** | **0.9134** | 1.4999 | 0.9505 | 1.6004 | 0.9952 | 1.6009 | 0.9603 | 1.5128 | 0.9438 | 1.4967 | 0.9347 | 1.5919 | 0.9914 |
| ETTh2 | 96 | **0.8210** | **0.6895** | 0.9951 | 0.7847 | 1.2858 | 0.8875 | 1.0349 | 0.7879 | 0.9803 | 0.7703 | 0.9616 | 0.7644 | 1.0586 | 0.8083 |
| exchange-rate | 96 | **1.8774** | **0.7607** | 3.1668 | 1.1146 | 4.4656 | 1.4831 | 2.2656 | 1.0016 | 2.6426 | 0.9977 | 2.6365 | 1.0061 | 3.0564 | 1.1111 |
| stock | 21 | **0.3239** | **0.3695** | 0.7554 | 0.6330 | 1.1194 | 0.8057 | 0.4936 | 0.4755 | 0.5135 | 0.4926 | 0.5256 | 0.5038 | 0.4900 | 0.4866 |

## 4.3 EVALUATION ON ZERO-SHOT SETTING

**Setups.** In this section, we delve into zero-shot learning to evaluate the transferability of Dual-Forecaster from the source domains to the target domains. In this setting, models trained on one dataset ♢ are evaluated on another dataset ♡, where the model has not encountered any data samples from the dataset ♡. We adopt the ETT datasets testing cross-domain adaptation.

**Results.** As shown in Table 3, in most cases, our approach consistently outperforms the most competitive baselines, which have leading performance on synthetic dataset and captioned-public datasets, surpassing iTransformer by **7.9%/5.6%** w.r.t. MSE/MAE. We note that Dual-Forecaster yields significantly better results with an average enhancement of **13.4%/9.1%** w.r.t. MSE/MAE compared to MM-TFSlib. We attribute this improvement to our model's superior ability to capture general knowledge about the dynamics of time series patterns through its sophisticated multimodal comprehension capabilty, enabling it to adapt to new datasets with this information during time series forecasting.

## 4.4 MODEL ANALYSIS

**Cross-modality Alignment.** To better illustrate the effectiveness of the model design in Dual-Forecaster, we construct 6 model variants and conduct ablation experiments on synthetic dataset and ETTm2 dataset. The experimental results presented in Table 4 demonstrate the importance of integrating both descriptively historical textual information and predictive textual insights for time

Table 3: Zero-shot learning results on ETT datasets in predicting 96 steps ahead. $\diamondsuit \rightarrow \heartsuit$ indicates that models trained on the dateset $\diamondsuit$ are evaluated on an entirely different dataset $\heartsuit$. **Red**: the best, Blue: the second best.

| Methods | Dual-Forecaster | | PatchTST | | iTransformer | | MM-TSFlib | |
|---|---|---|---|---|---|---|---|---|
| Metric | MSE | MAE | MSE | MAE | MSE | MAE | MSE | MAE |
| ETTm1 → ETTm2 | 1.2094 | 0.7239 | 1.3132 | 0.7750 | **1.1627** | **0.7219** | 1.2026 | 0.7427 |
| ETTm1 → ETTh2 | **1.4103** | **0.9230** | 1.7364 | 1.0224 | 1.6858 | 1.0172 | 1.8196 | 1.0615 |
| ETTm2 → ETTm1 | **1.4029** | **0.8282** | 2.4748 | 1.0873 | 1.4702 | 0.8643 | 1.5251 | 0.8853 |
| ETTm2 → ETTh2 | **1.8655** | **1.0657** | 2.5812 | 1.2549 | 2.1917 | 1.1737 | 2.4401 | 1.2423 |

Table 4: Ablation on synthetic dataset and ETTm2 with prediction horizon 30 and 96, respectively. The best results are highlighted in **bold**.

| Model Variants | synthetic dataset | | ETTm2 | |
|---|---|---|---|---|
| | MSE | MAE | MSE | MAE |
| **Dual-Forecaster** | **0.5150** | **0.4703** | **0.8469** | **0.5756** |
| **w/o Any Texts** | 0.6518 | 0.5593 | 0.9507 | 0.6060 |
| **History Texts Injection** | **0.6313** | **0.5458** | **0.9363** | **0.6108** |
| → w/o History Text-Time Series Contrastive Loss | 0.6868 | 0.5794 | 0.9571 | 0.6117 |
| → w/o History-oriented Modality Interaction Module | 0.6418 | 0.5502 | 0.9480 | 0.6102 |
| **Future Texts Injection** | **0.5150** | **0.4703** | **0.8469** | **0.5756** |
| → w/o Future-oriented Modality Interaction Module | 0.6313 | 0.5458 | 0.9363 | 0.6108 |
| → w/o History Texts Injection | 0.5361 | 0.4833 | 0.9105 | 0.5907 |

series forecasting to achieve optimal performance, and also validate the soundness of the design of three cross-modality alignment techniques. Employing only historical textual information results in MSE/MAE of **0.6313/0.5458** on synthetic dataset and **0.9363/0.6108** on ETTm2, respectively. Without History-oriented Modality Interaction Module, we observe an average performance degradation of **0.9%**, while the average performance reduction becomes more obvious (**4.3%**) in the absence of History Text-Time Series Contrastive Loss. Thanks to the design of Future-oriented Modality Interaction Module, the addition of future textual insights leads to pronounced improvements (over **14.0%/9.8%** w.r.t. MSE/MAE reduction), achieving the lowest MSE and MAE on both datasets. It is worth mentioning that relying solely on future textual insights, devoid of history texts, fails to achieve optimal forecasting performance, thereby proving the significance of concurrently integrating both history and future texts.

**Cross-modality Alignment Interpretation.** We present a case study on synthetic dataset, as illustrated in Figure 3, to demonstrate the alignment effect between text and time series. This is achieved by displaying the similarity matrix that captures the relationship between text features and time series embeddings. The time series data is visualized above the matrix, while its corresponding text descriptions are on the left. This result shows that Dual-Forecaster is capable of autonomously discern potential connections between text and time series except be able to accurately recognize the genuine pairing text-time series relationships. This indicates that our model possesses advanced multimodal comprehension capability, which has a positive influence on improving the model's forecasting performance.

**Cross-modality Alignment Efficiency** Table 5 provides an overall efficiency analysis of Dual-Forecaster with and without cross-modal alignment techniques. Our model's unimodal time series encoder is lightweight, and the overall efficiency of Dual-Forecaster is actually capped by the leveraged effective cross-modal alignment module. This is favorable in balancing forecating performance and efficiency.

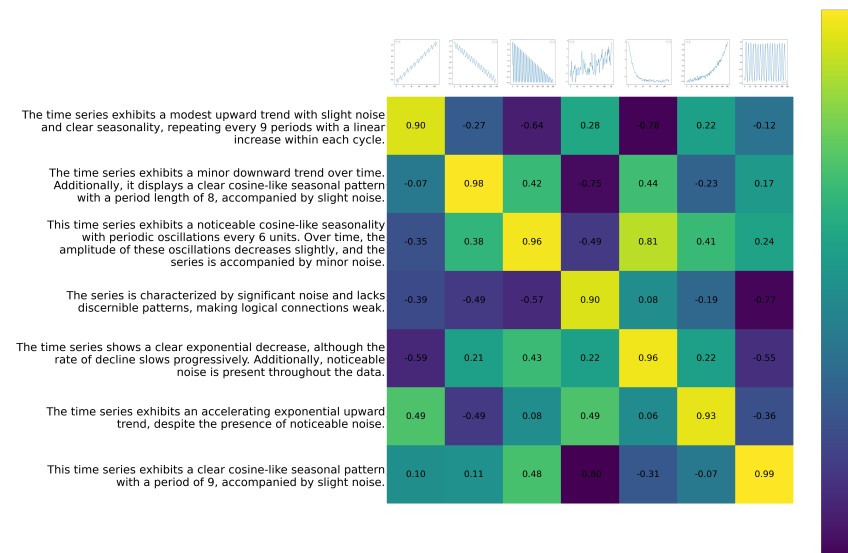

Figure 3: A showcase of text-time series alignment. The values in the matrix represent the similarity between the high-dimensional representation of the time series (above the matrix) and the corresponding textual description (on the left side of the matrix). The higher the similarity, the better the match between the time series and the text.

Table 5: Efficiency analysis of Dual-Forecaster on synthetic dataset and ETTm2.

| Dataset-Prediction Horizon | synthetic dataset-30 | | | | ETTm2-96 | | | |
|---|---|---|---|---|---|---|---|---|
| Metric | Trainable Param. (M) | Non-trainable Param. (M) | Mem. (MiB) | Speed(s/iter) | Trainable Param. (M) | Non-trainable Param. (M) | Mem. (MiB) | Speed(s/iter) |
| Future Texts Injection (*Modality Interaction II* in Figure 1) | 14.6 | 82.1 | 1852 | 0.068 | 14.6 | 82.1 | 8918 | 0.448 |
| History Texts Injection (*Modality Interaction I* in Figure 1) | 13.5 | 82.1 | 1840 | 0.043 | 13.6 | 82.1 | 8812 | 0.242 |
| w/o Any Texts (Unimodal Time Series Encoder in Figure 1) | 6.5 | 0 | 672 | 0.022 | 6.6 | 0 | 928 | 0.036 |

## 5 CONCLUSION

In this work, we present Dual-Forecaster, an innovative multimodal time series model aiming at generating more reasonable forecasts. It is augmented by rich, descriptively historical textual information and predictive textual insights, all supported by advanced multimodal comprehension capability. To enhance its multimodal comprehension capability, we craft three cross-modality alignment techniques, including historical text-time series contrastive loss, history-oriented modality interaction module, and future-oriented modality interaction module. We conduct extensive experiments on synthetic dataset and captioned-public datasets to demonstrate the effectiveness of Dual-Forecaster and the superiority of integrating textual data for time series forecasting.

**Limitations & Future Work** While Dual-Forecaster has achieved remarkable performance in text-guided time series forecasting, there remains room for further improvements. Due to resource constraints, a comprehensive hyperparameter tuning was not performed, suggesting that the reported results of Dual-Forecaster may be sub-optimal. In terms of multimodal time series dataset, the lack of a standardized and efficient annotation methodology often leads to inadequate annotation quality on real-world datasets, with the issue being particularly pronounced in the annotation of long time series. Future work should focus on developing a more elegant time series annotator, leveraging the text-time series alignment techniques that are fundamental to Dual-Forecaster. In terms of downstream task, further research should explore the potential of expanding Dual-Forecaster to encompass a broad spectrum of multimodal time series analysis capabilities.

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

## A  IMPACT STATEMENT

This paper presents work whose goal is to advance the field of Machine Learning. There are many potential societal consequences of our work, none which we feel must be specifically highlighted here.

## B  EXPERIMENTAL DETAILS

### B.1  IMPLEMENTATION

All the experiments are repeated three times with different seeds and we report the averaged results. Our model implementation is on Pytorch (Paszke et al., 2019) with all experiments conducted on a single NVIDIA GeForce RTX 4070 Ti GPU. Our detailed model configurations are in Appendix B.4.

### B.2  MULTIMODAL TIME SERIES BENCHMARK DATASETS CONSTRUCTION

In the realm of time series forecasting, there is a notable lack of high-quality multimodal time series benchmark datasets that combine time series data with corresponding textual series. While some studies have introduced multimodal benchmark datasets (Liu et al., 2024b; Xu et al., 2024), these datasets primarily rely on textual descriptions derived from external sources like news reports or background information. These types of textual data are often domain-specific and may not be consistently available across different time series domains, limiting their utility for building unified multimodal models. In contrast, shape-based textual descriptions of time series patterns are relatively easier to generate and can provide more structured insights. The TS-Insights dataset Zhang et al. (2023) pairs time series data with shape-based textual descriptions. However, these descriptions are based on detrended series (with seasonality removed), which may introduce bias and complicate the interpretation of the original time series data. To address these challenges, we propose five new multimodal time series benchmark datasets where textual descriptions are directly aligned with the observed patterns in the time series. The construction process for these datasets is outlined below.

#### B.2.1  SYNTHETIC DATASET

For the synthetic time series data, we firstly design three categories of components, which are then combined to generate simulated time series. The components are as follows:

- **Trend:** Linear trend, exponential trend
- **Seasonality:** Cosine, linear, exponential, M-shape, trapezoidal
- **Noise:** Gaussian noise with varying variances

To generate the synthetic time series, one component from each category is randomly selected. These components are then either added together or multiplied to produce a time series, along with a corresponding textual description of its key characteristics. To enhance the diversity of the descriptions, rule-based descriptions are paraphrased using GPT-4o. Additionally, to simulate transitions between different states, we generate time series where only one component changes over time. For instance, a time series might exhibit a linear upward trend that transits to a linear downward trend. In this manner, we construct the synthetic dataset with a total of 3,040 training samples. Each sample includes historical time series and future time series, as well as paired historical and future textual series. Several examples of these constructed samples are shown in Figure 4.

#### B.2.2  CAPTIONED PUBLIC DATASETS

For the real-world time series data, we construct corresponding textual descriptions using the following method, and figure 5 shows the whole caption process.

- First, we apply the Iterative End Point Fitting (IEPF) algorithm (Douglas & Peucker, 1973) to the min-max normalized time series, identifying reasonable segmentation points. IEPF begins by taking the starting curve, which consists of an ordered set of points, and an allowable distance threshold. Initially, the first and last points of the curve are marked as

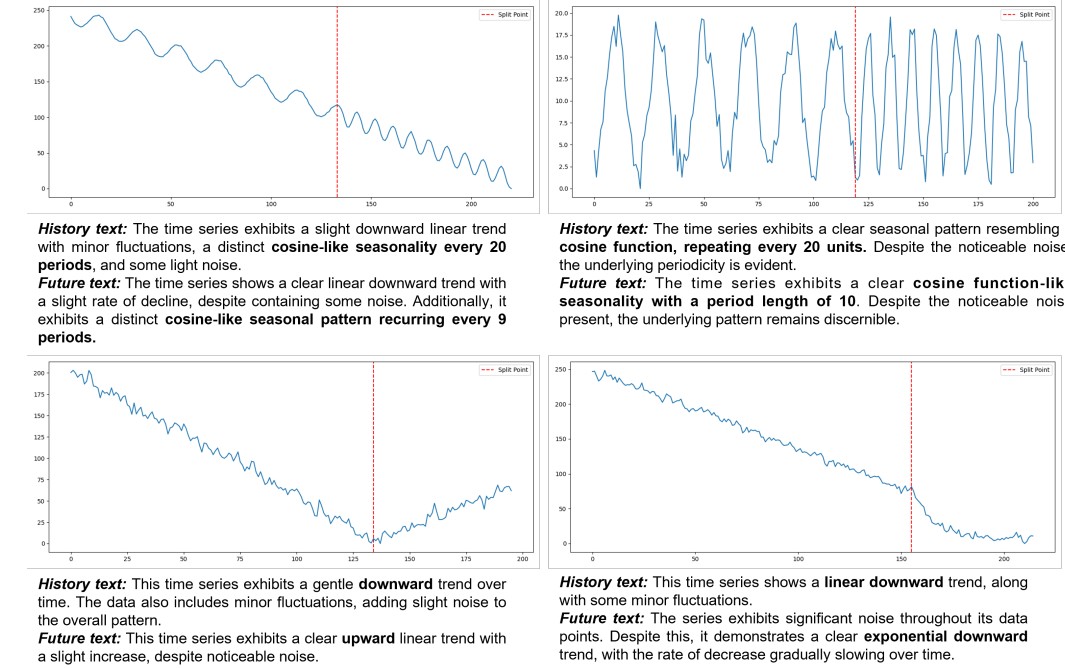

**History text:** The time series exhibits a slight downward linear trend with minor fluctuations, a distinct **cosine-like seasonality every 20 periods**, and some light noise.
**Future text:** The time series shows a clear linear downward trend with a slight rate of decline, despite containing some noise. Additionally, it exhibits a distinct **cosine-like seasonal pattern recurring every 9 periods.**

**History text:** The time series exhibits a clear seasonal pattern resembling a **cosine function, repeating every 20 units.** Despite the noticeable noise, the underlying periodicity is evident.
**Future text:** The time series exhibits a clear **cosine function-like seasonality with a period length of 10**. Despite the noticeable noise present, the underlying pattern remains discernible.

**History text:** This time series exhibits a gentle **downward** trend over time. The data also includes minor fluctuations, adding slight noise to the overall pattern.
**Future text:** This time series exhibits a clear **upward** linear trend with a slight increase, despite noticeable noise.

**History text:** This time series shows a **linear downward** trend, along with some minor fluctuations.
**Future text:** The series exhibits significant noise throughout its data points. Despite this, it demonstrates a clear **exponential downward** trend, with the rate of decrease gradually slowing over time.

Figure 4: Synthetic time series and its paired text examples.

essential. The algorithm then iteratively identifies the point farthest from the line segment connecting these endpoints. If the distance of this point exceeds threshold, it is retained as a segmentation point, and the process is recursively repeated for the subsegments until no points are found that are farther than threshold from their respective line segments. This iterative approach ensures that the segmentation preserves the curve's critical structure while discarding unnecessary details. The lines connecting these segmentation points can roughly outline the overall shape of the time series.

- Once the time series is segmented, statistical features such as slope and volatility are computed for each section. For each segment, a linear regression model is fitted to the data, and the slope is calculated. The P-value from the regression determines the significance of the trend: if it's below 0.05, the slope indicates an upward or downward trend; if it's above 0.05, the segment is considered to be fluctuating. The Mean Squared Error (MSE) between the original data and the regression line is also calculated to measure the noise level. Based on the MSE, the noise is classified into three levels: low, medium, or high.

- Finally, a textual description is generated: if the local trend is significant, the description notes whether the segment is increasing or decreasing; if not, it indicates fluctuation. The noise level is also included in the description based on the MSE.

We apply the above method to annotate six commonly used real-world datasets: ETTm1, ETTm2, ETTh1, ETTh2, exchange-rate, and stock. Each dataset is divided into training and testing sets with a ratio of 8:2. Following the configuration of a look-back window of 336 and a forecasting horizon of 96, we construct training samples using a sliding window approach. For each training instance, both the historical and future time series are annotated. Figure 6 and figure 7 illustrate the text annotation results on two real-world datasets. Our annotation method accurately captures segmentation points (red lines), thereby producing meaningful summary shape descriptions.

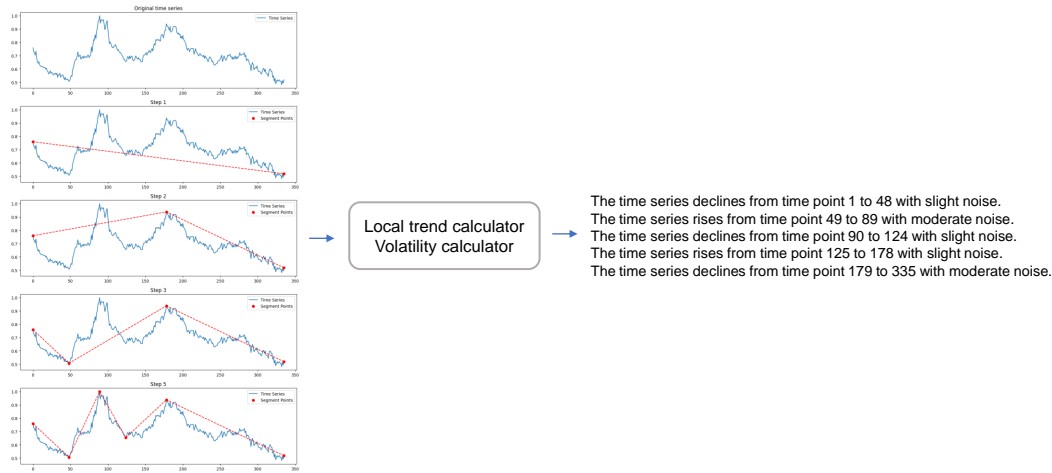

Figure 5: Captioning process for real-world time series. First, IEPF is used to segment time series, identifying reasonable segmentation points. This algorithm works by iteratively fitting straight lines between endpoints and adjusting segmentation points to minimize fitting errors, thereby identifying rational breakpoints. Next, statistical features such as slope and volatility are calculated for each segmented portion of the time series. Finally, based on these statistical characteristics, a descriptive textual summary is generated.

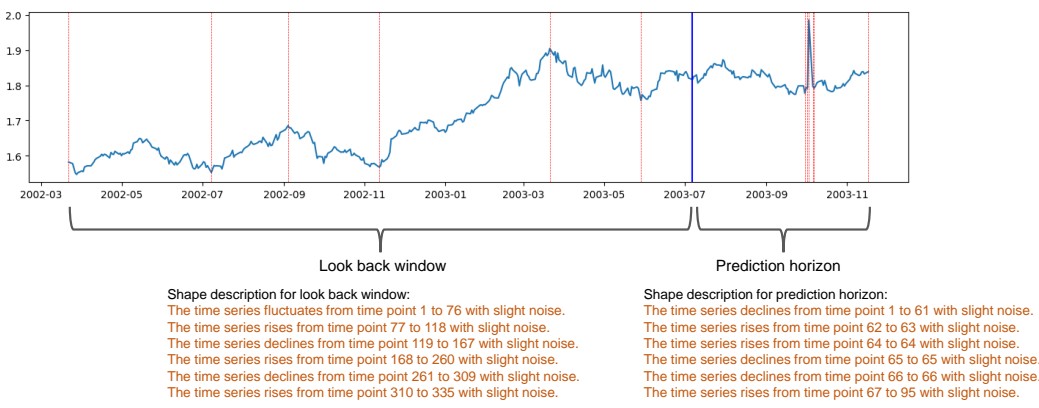

Figure 6: Visualization of a captioned example from exchange-rate dataset.

## B.3 EVALUATION METRIC

We adopt the Mean Square Error (MSE) and Mean Absolute Error (MAE) as the default evaluation metrics. The calculations of these metrics are as follows:

$$MSE = \frac{1}{H} \sum_{h=1}^{H} \left( Y_h - \hat{Y}_h \right)^2$$

$$MAE = \frac{1}{H} \sum_{h=1}^{H} \left| Y_h - \hat{Y}_h \right|$$

where $H$ denotes the length of prediction horizon. $Y_h$ and $\hat{Y}_h$ are the $h$-th ground truth and prediction where $h \in \{1, \cdots, H\}$.

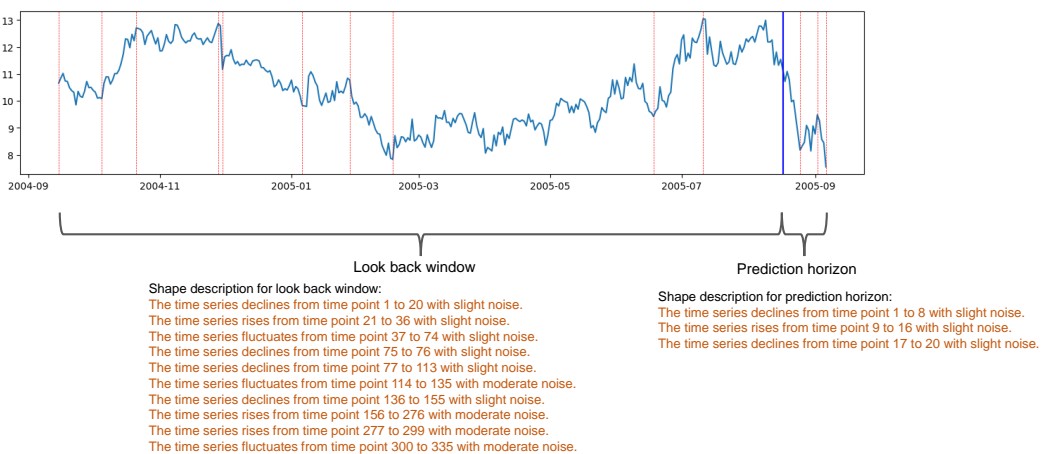

Figure 7: Visualization of a captioned example from stock dataset.

## B.4 MODEL CONFIGURATIONS

The configurations of our models, in relation to the evaluations on various datasets, are consolidated in Table 6. By default, optimization is achieved through the Adam optimizer (Kingma, 2014) with a learning rate set at 0.0001 and a weight deacy ratio of 0.01, throughout all experiments. We utilize six-layers *RoBERTa* (Liu, 2019) to process text inputs. In terms of dataset parameter, *L* and *h* signify the input time series *look back window* length and the future time points to be predicted, respectively. For the input time series, we first perform patching to obtain *P* non-overlapping patches with a patch length of $L_p$. In terms of model hyperparameter, $d_m$ represents the dimension of the embedded representations. $n_{uni}$ denotes the number of layers of unimodal time series encoder used to process time series inputs, while $n_{mul}$ denotes the number of layers of the history-oriented modality interaction module, which ensures effectively alignment of distributions between historical textual and time series data. Heads are correlate to the *Multi-Head Self-Attention* (*MHSA*) and *Multi-Head Cross-Attention* (*MHCA*) operations utilized for cross-modality alignment. For the synthetic dataset, we set the training epochs to 300, while for the ETT, exchange-rate and stock dataset, we set it to 100. Additionally, to prevent overfitting, we introduce an early stopping strategy and set the patience to 7.

Table 6: An overview of the experimental configurations for Dual-Forecaster.

| Dataset/Configuration | Dataset Parameter | | | | Model Hyperparameter | | | | Training Process | | | | |
|---|---|---|---|---|---|---|---|---|---|---|---|---|---|
| | *L* | *P* | $L_p$ | *h* | $d_m$ | $n_{uni}$ | $n_{mul}$ | Heads | LR | Weight Decay | Batch Size | Epochs | Patience |
| synthetic dataset | 200 | 25 | 8 | 30 | 256 | 6 | 3 | 8 | 0.0001 | 0.01 | 64 | 300 | 7 |
| ETTm1 | 336 | 42 | 8 | 96 | 256 | 6 | 3 | 8 | 0.0001 | 0.01 | 64 | 100 | 7 |
| ETTm2 | 336 | 42 | 8 | 96 | 256 | 6 | 3 | 8 | 0.0001 | 0.01 | 64 | 100 | 7 |
| ETTh1 | 336 | 42 | 8 | 96 | 256 | 6 | 3 | 8 | 0.0001 | 0.01 | 64 | 100 | 7 |
| ETTh2 | 336 | 42 | 8 | 96 | 256 | 6 | 3 | 8 | 0.0001 | 0.01 | 64 | 100 | 7 |
| exchange-rate | 336 | 42 | 8 | 96 | 256 | 6 | 3 | 8 | 0.0001 | 0.01 | 64 | 100 | 7 |
| stock | 336 | 42 | 8 | 21 | 256 | 6 | 3 | 8 | 0.0001 | 0.01 | 64 | 100 | 7 |

## C BASELINES

**DLinear:** a combination of a decomposition scheme and a linear network that first divides a time series data into two components of trend and remainder, and then performs forecasting to the two series respectively with two one-layer linear model.

**FITS:** consists of the key part of the complex-valued linear layer that is dedicatedly designed to learn amplitude scaling and phase shifting, thereby facilitating to extend time series segment by interpolating the frequency representation.

**PatchTST:** is composed of two key components: (i) patching that segments time series into patches as input tokens to Transformer; (ii) channel-independent structure where each channel univariate time series shares the same Transformer backbone.

**iTransformer:** is an inverted Transformer that raw series of different variates are firstly embedded to tokens, applied by self-attention for multivariate correlations, and individually processed by the share feed-forward network for series representations of each token.

**MM-TSFlib:** is the first multimodal time-series forecasting (TSF) library, which allows the integration of any open-source language models with arbitrary TSF models, thereby enabling multimodal TSF tasks based on Time-MMD.

**Time-LLM:** is a new framework, which encompasses reprogramming time series data into text prototype representations before feeding it into the frozen LLM and providing input context with declarative prompts via Prompt-as-Prefix to augment reasoning.

## D ERROR BARS

All experiments have been conducted three times, and we present the average MSE and MAE, as well as standard deviations here. The comparison between our method and the second-best method, PatchTST (Nie et al., 2022), on synthetic dataset, are delineated in Table 7. Furthermore, Table 8 contrasts the effectiveness of our method with that of the runner-up method, MM-TSFlib (Liu et al., 2024b), on captioned-public datasets.

Table 7: Standard deviations of our Dual-Forecaster and the second-best method (PatchTST) on synthetic dataset.

| Model | Dual-Forecaster | | PatchTST | |
|---|---|---|---|---|
| Dataset | MSE | MAE | MSE | MAE |
| synthetic dataset | 0.5150±0.0209 | 0.4703±0.0119 | 0.6015±0.0063 | 0.5394±0.0057 |

Table 8: Standard deviations of our approach and the runner-up method (MM-TSFlib) on ETT, exchange-rate and stock datasets.

| Model | Dual-Forecaster | | MM-TSFlib | |
|---|---|---|---|---|
| Dataset | MSE | MAE | MSE | MAE |
| ETTm1 | 1.2126±0.0161 | 0.7686±0.0088 | 1.3620±0.0085 | 0.8426±0.0101 |
| ETTm2 | 0.8469±0.0222 | 0.5756±0.0090 | 1.0325±0.0157 | 0.6691±0.0025 |
| ETTh1 | 1.4190±0.0154 | 0.9134±0.0054 | 1.4967±0.0333 | 0.9347±0.0135 |
| ETTh2 | 0.8210±0.0173 | 0.6895±0.0098 | 0.9616±0.0080 | 0.7644±0.0021 |
| exchange-rate | 1.8774±0.0276 | 0.7607±0.0182 | 2.6365±0.0958 | 1.0061±0.0275 |
| stock | 0.3239±0.0050 | 0.3695±0.0019 | 0.5256±0.0114 | 0.5038±0.0098 |

## E VISUALIZATION

In this part, we visualize the forecasting results of Dual-Forecaster compared with the state-of-the-art and representative models (*e.g.*, MM-TSFlib (Liu et al., 2024b), iTransformer (Liu et al., 2023b), PatchTST (Nie et al., 2022), Time-LLM(Jin et al., 2023)) in various scenarios to demonstrate the superior performance of Dual-Forecaster. Among the various models, Dual-Forecaster predicts the most precise future series variations and displays superior performance.

Moreover, we provide a case visualization from ETTm2 dataset to elucidate how future texts guide the forecasting process. It demonstrates that our model is adept at effectively comprehending the pivotal insights (*e.g.*, exact time points and trend information) embedded within future textual data. Owing to this information, our model is empowered to produce more reasonable forecasts.

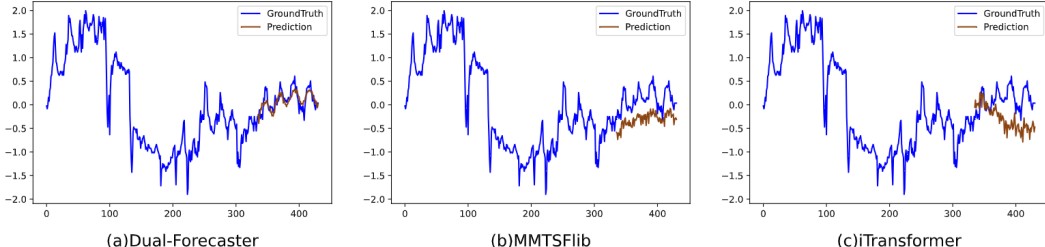

Figure 8: Visualization of an example from ETTh1 under the input-336-predict-96 settings. Blue lines are the ground truths and brown lines are the model predictions.

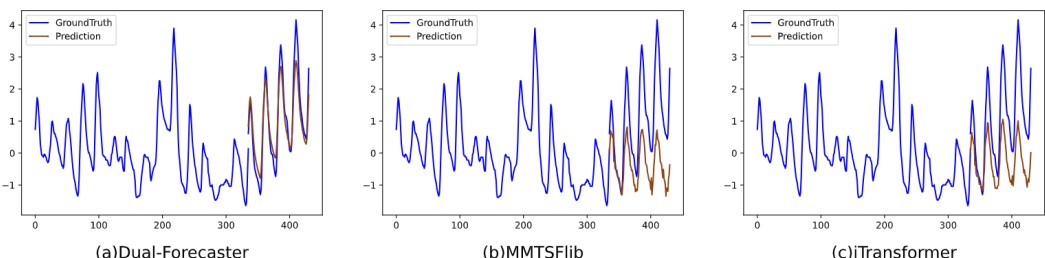

Figure 9: Visualization of an example from ETTh2 under the input-336-predict-96 settings. Blue lines are the ground truths and brown lines are the model predictions.

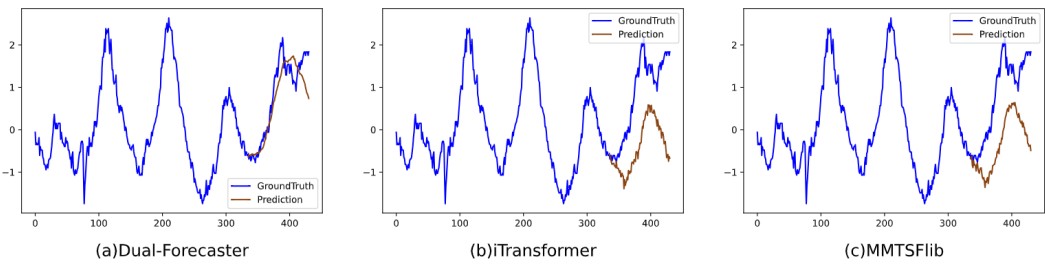

Figure 10: Visualization of an example from ETTm1 under the input-336-predict-96 settings. Blue lines are the ground truths and brown lines are the model predictions.

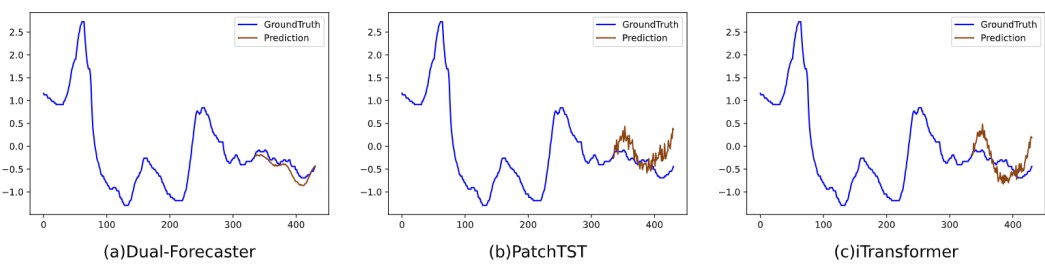

Figure 11: Visualization of an example from ETTm2 under the input-336-predict-96 settings. Blue lines are the ground truths and brown lines are the model predictions.

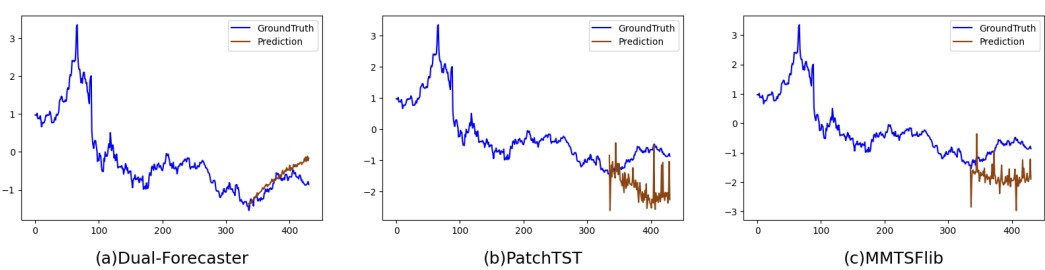

Figure 12: Visualization of an example from exchange-rate dataset under the input-336-predict-96 settings. Blue lines are the ground truths and brown lines are the model predictions.

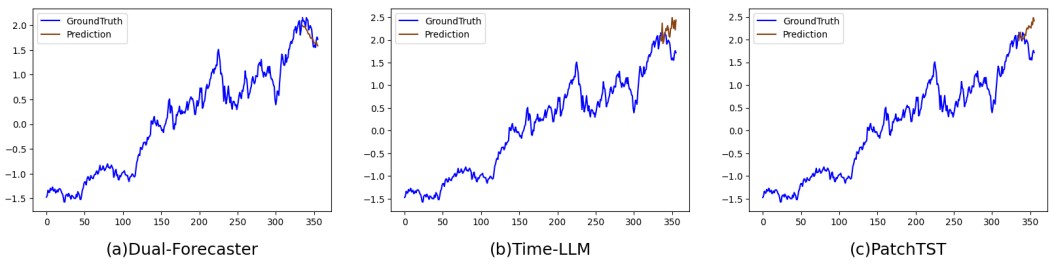

Figure 13: Visualization of an example from stock dataset under the input-336-predict-21 settings. Blue lines are the ground truths and brown lines are the model predictions.

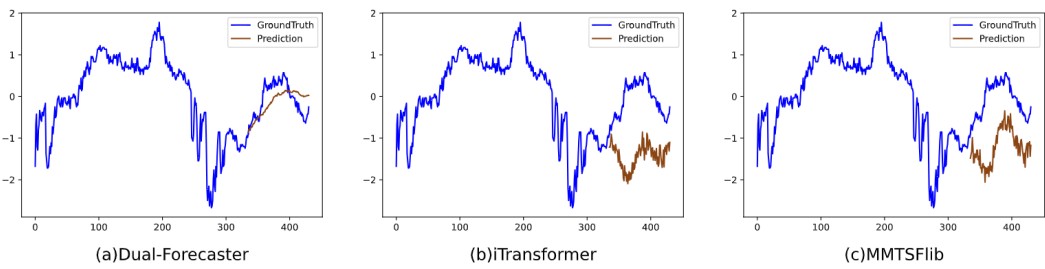

Figure 14: Zero-shot forecasting case from ETTm2 → ETTm1 under the input-336-predict-96 settings. Blue lines are the ground truths and brown lines are the model predictions.

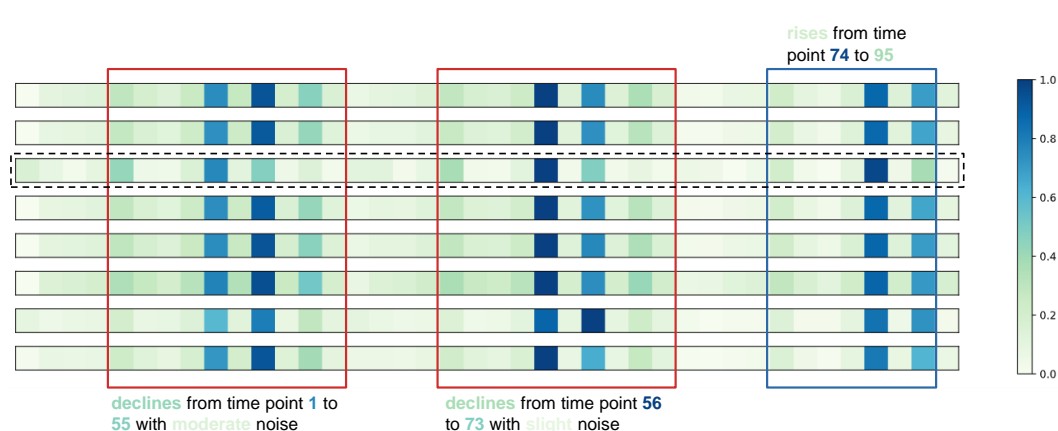

Figure 15: Attention map visualizing the relationship between the last time series token embedding and future text tokens. Eight subplots correspond to the 8 heads of *Multi-Heads Cross-Attention* operation, respectively. Case comes from the ETTm2 dataset.

Table 9: Ablation study results of different pre-trained models on ETTm2 dataset.

|  |  | *RoBERTa* |  | GPT2 |  |
| --- | --- | --- | --- | --- | --- |
| Metric |  | MSE | MAE | MSE | MAE |
| ETTm2 | 96 | 0.8469 | 0.5756 | 0.8522 | 0.5749 |

Table 10: Forecasting result on traffic and weather datasets. The best result is highlighted in **bold** and the second best is highlighted in underlined.

| Methods |  | Dual-Forecaster |  | DLinear |  | FITS |  | PatchTST |  | iTransformer |  | MM-TSFlib |  | Time-LLM |  |
| --- | --- | --- | --- | --- | --- | --- | --- | --- | --- | --- | --- | --- | --- | --- | --- |
| Metric |  | MSE | MAE | MSE | MAE | MSE | MAE | MSE | MAE | MSE | MAE | MSE | MAE | MSE | MAE |
| traffic | 24 | **0.2517** | **0.2254** | 0.3883 | 0.4037 | 0.5450 | 0.5166 | 0.2592 | 0.2781 | 0.2587 | 0.2775 | 0.2731 | 0.2984 | 0.3521 | 0.3585 |
| weather | 30 | **1.6310** | **0.3980** | 1.7508 | 0.6126 | 1.9136 | 0.6643 | 1.7711 | 0.6320 | 1.6471 | 0.5519 | 1.6467 | 0.5566 | 1.6160 | 0.5063 |

