# OpenReview forum: "Dual-Forecaster: A Multimodal Time Series Model Integrating Descriptive and Predictive Texts"
_ICLR.cc/2025/Conference — ICLR 2025 Conference Withdrawn Submission_

### Official Review · Reviewer_cjGf · 2024-10-23

**Soundness:** 3
**Presentation:** 3
**Contribution:** 3
**Rating:** 5
**Confidence:** 5

**Summary:**

Based on transformer, this paper introduces Dual-Forecaster, a multimodal time series forecasting model designed to enhance prediction accuracy by integrating both numerical time series data and corresponding textual information, where the data is used and aligned to help time-series prediction task. To effectively merge these modalities, Dual-Forecaster incorporates three cross-modality alignment techniques: Historical Text-Time Series Contrastive Loss, History-oriented Modality Interaction Module, Future-oriented Modality Interaction Module.

**Strengths:**

1. Point out the information insufficiency problem, which I agree.
2. The model is clean and simple, and it works well in synthetic data. The use of pre-trained model and its application is acceptable.

**Weaknesses:**

1. I wonder how to use the model as the textual data is well-designed, and it seems that it performs bad when I don't input. For most of the users, I just wanna use the model as quick as possible.
2. What about I use wrong information in texts, this interests me.
3. The example is too simple in Figure2
4. Compare with some multi-modal approach, (TIME-LLM eg.). the comparison is weak.
5. give more ablation study including pre-trained model.

**Questions:**

1. explain why textual data, in particular, is the most effective supplementary modality
2. why use the contrastive loss ?

---

> ### Author Response · Authors · 2024-11-21
>
> Thanks for careful reading and the useful suggestions. Here we would like to respond to the weaknesses and questions you mentioned. Hope your concerns will be successfully addressed.
>
> ## **Response to Weakness #1** ##
>
> **We have to emphasize that the motivation of our research work is to integrate textual data to address the challenges posed by insufficient information in single-modal models, and thus we propose Dual-Forecaster that aims to effectively utilize textual data to assist time series data for more accurate forecasting.** Ablation experimental results shown in Table 4 indicate that incorporating descriptive historical textual data can enhance the forecasting performance compared to relying purely on time series data. When further combining predictive textual data, the forecasting performance will be further improved. It demonstrates the significance of incorporating textual data and the effectiveness of our model in utilizing textual data for improving forecasting performance. Additionally, the comparative experimental results with the two multimodal methods of MM-TFSlib and Time-LLM highlight the superiority of our model in using textual data to assist time series data for forecasting. We acknowledge that without any text input, our model's forecasting performance cannot consistently outperform other single-modal baseline models, e.g., DLinear, FITS, PatchTST, and iTransformer, but our model always ranks among the top (the count of the best and the second best results is 4 and 2, respectively, including synthetic dataset and captioned-ETTm1/ETTm2/ETTH1/ETTH2/exchange-rate/stock datasets, where the exchange-rate and stock datasets are the two newly added datasets according to the other reviewer’s suggestions). Thank you for your recognition of our model and your expressed willingness to use it as quick as possible. Our model adopts a modular design, which can quickly adapt to forecasting scenarios with or without text input. Moreover, our model can ensure favorable forecasting performance even without any text input.
>
> ## **Response to Weakness #2** ##
>
> This is a common response raised by reviewers, so we provide a unified reply. Please refer to “**Misleading Textual Data**” section in the Common Response.
>
> ## **Response to Weakness #3** ##
>
> The showcase in Figure 2 is from the synthetic dataset, which is intended to **more intuitively illustrate** the importance of incorporating textual data to improve forecasting performance. As described in the construction method of the synthetic dataset in Appendix B.2.1, examples containing state changes **are relatively more difficult** to forecast in the synthetic dataset. Moreover, in Appendix E, we also provide visualizations of our model and other SOTA models' forecasts on captioned-public datasets (Figures 8 to Figure 13), demonstrating that our model can better utilize textual data and provide more reasonable forecasts.
>
> ## **Response to Weakness #4** ##
>
> **From the comparative experimental results shown in Table 1 and Table 2, it can be seen that we have chosen two multimodal methods, Time-LLM and MM-TFSlib.** As mentioned in the second part of RELATED WORK, there is currently limited relevant research on multimodal time series forecasting, including Time-LLM, MM-TFSlib, TGForecaster, and TimeCMA. Among them, TGForecaster and TimeCMA do not release code, and the textual data used by TGForecaster is the channel description and news corresponding to each future time stamps, while the textual data our model use is the overall shape-based descriptions of time series patterns within the corresponding time period. Due to the inability to make a fair comparison with them, we do not choose these two multimodal methods as baselines. Therefore, we finally choose Time-LLM and MM-TFSlib as baselines.
>
> ## **Response to Weakness #5** ##
>
> Thank you for pointing out this issue. We **supplement the ablation study** to evaluate the impact of different language pre-trained models (RoBERTa and GPT2) on the forecasting performance of Dual-Forecaster on ETTm2 dataset. The experimental results have been added in the revised paper (see Table 9 in Appendix, marked in Green).
>
> If we misunderstand your meaning, please let us know. We will supplement the corresponding experiments as quick as possible to address your concerns.

---

> > ### Author Response · Authors · 2024-11-21
> >
> > ## **Response to Question #1** ##
> >
> > **We have to clarify in advance that we do not make a conclusion on which supplementary modality is most effective for time series forecasting.** Whether textual data is the most effective supplementary modality requires more rigorous experiments for verification. We use textual data as a supplementary modality mainly for the following two reasons:
> > 1. In real-world time series forecasting scenarios, textual data is the most common type of supplementary modality, such as news, expert knowledge, etc., which usually exist in text form. Therefore, studying how textual data can assist time series forecasting is more practically significant.
> > 2. Textual data and time series data all belong to one-dimensional sequence data, and their modeling methods are relatively similar, making them easier for unified modeling.
> >
> > Experimental results demonstrate that utilizing textual data as supplementary modality is effective in improving our model’s forecasting performance.
> >
> > ## **Response to Question #2** ##
> >
> > Textual data and time series data belong to two independent types of modality data, with their respective representations located in different high-dimensional spaces. **To effectively integrate the two modalities, enable the model to better align them and learn the relationships between different variables in time series data, we design the Historical Text-Time Series Contrastive Loss.** Its core idea is to narrow the distance between text representation and time series representation in high-dimensional space, that is to convert text into a "language" that are more naturally suited to time series models for maximizing the utilization of textual data to obtain high-quality time series representations. As shown in Figure 3, after using contrastive loss, the textual data and time series data can be accurately aligned, with significantly higher values on the diagonal than on the non-diagonal. For example, the model can accurately distinguish the time series patterns corresponding to "downward trend" and "cosine-like seasonal patterns" (see small figures 2 and 7 above the matrix, numbered from left to right), and the similarity values of text representation and time series representation are 0.98 and 0.99, respectively, significantly higher than those at other positions. Additionally, the model can automatically discern potential relationships between different variables. Taking the fifth column of the matrix as an example, it is not difficult to see that for the small figure 5 above the matrix, its corresponding time series representation is highly like the genuine pairing text representation (similarity value of 0.96). Moreover, it also has varying degrees of similarity with the text representations corresponding to small figures 3 and 2 (0.81 and 0.44), which indicates that the model can utilize textual data to learn the correlations between different variables, making the learned time series representations have richer information and thus improving the quality of time series representations. The effectiveness of using contrastive loss is validated from the ablation experimental results in Table 3 (see the row of "→ w/o History Text Time Series Contrastive Loss").

---

> > > ### Comment · Reviewer_cjGf · 2024-11-21
> > >
> > > Thank you for your quick reply,
> > > please also test on Traffic and Weather dataset. ETT series and exchange rate is relatively simple.
> > > It's acceptable if the performance is not good, but please give more insights about why it failed and how would you improve.

---

> > > > ### Author Response · Authors · 2024-11-22
> > > >
> > > > Thanks for your quick response.
> > > >
> > > > We **have added comparative experiments on Traffic and Weather datasets** (see Table 10 in Appendix, marked in Green). The comparative experimental results demonstrate that our model still outperforms other SOTA models on these two datasets.

---

> > > > > ### Author Response · Authors · 2024-11-25
> > > > >
> > > > > Dear Reviewer cjGf,
> > > > >
> > > > > We hope that our rebuttal, along with the modifications to the draft, have effectively addressed the weaknesses and questions raised in your review. We are eager to engage in further  questions or concerns regarding our work during the discussion period. If our clarifications and the newly added results align with your expectations, we respectfully request you to consider revising the score.

---

> > > > > > ### Author Response · Authors · 2024-11-29
> > > > > >
> > > > > > Dear Reviewer cjGf,
> > > > > >
> > > > > > We **have supplemented another comparative experiment** with all baselines (**including a new multi-modal approach of TGForecaster**) on the Weather-captioned dataset proposed in [1]. As described in [1], this dataset includes weather forecasting reports detailing the climate conditions, which are updated every six hours and daily. The authors employ GPT-4 with designed prompt to summarize these reports into seven brief sentences, which **include somehow wrong information**. This is also the part that you are interested in, mentioned in Weakness #2 and Weakness #4. Below shows the experimental results.
> > > > > >
> > > > > > ||Dual-Forecaster||TGForecaster||MM-TSFlib||Time-LLM||DLinear||FITS||PatchTST||iTransformer||
> > > > > > |:----|:----|:----|:----|:----|:----|:----|:----|:----|:----|:----|:----|:----|:----|:----|:----|:----|
> > > > > > |Metric|MSE|MAE|MSE|MAE|MSE|MAE|MSE|MAE|MSE|MAE|MSE|MAE|MSE|MAE|MSE|MAE|
> > > > > > |Weather_captioned|`0.0297`|`0.1174`|0.0600|0.1928|0.0575|0.1792|0.0449|0.1613|0.0677|0.1926|0.0628|0.1849|0.0525|0.1650|0.0558|0.1719|
> > > > > >
> > > > > > Experimental results show that Dual-Forecaster consistently outperforms other SOTA models, indicating that our model can still ensure excellent forecasting capability **even when the input text data contains wrong information**.
> > > > > >
> > > > > > We hope our responses could adequately address your concerns. And if so, whether the rating will be reevaluated to a higher level. Thank you very much!
> > > > > >
> > > > > > Sincerely,
> > > > > >
> > > > > > The Authors
> > > > > >
> > > > > > [1] TGTSF: Beyond Trend and Periodicity: Guide Time Series Forecasting with Textual Cues.

---

> > > > > > > ### Author Response · Authors · 2024-12-02
> > > > > > >
> > > > > > > Dear Reviewer cjGf,
> > > > > > >
> > > > > > > We hope the above responses have adequately addressed your concerns. We are eager to engage in further questions or concerns regarding our work during the discussion period. If our clarifications align with your expectations, we respectfully request you to consider revising the score.
> > > > > > >
> > > > > > > Best regards,
> > > > > > > Authors

---

### Official Review · Reviewer_E6mB · 2024-10-30

**Soundness:** 2
**Presentation:** 3
**Contribution:** 3
**Rating:** 5
**Confidence:** 3

**Summary:**

The paper introduces Dual-Forecaster, a novel multimodal time series forecasting model that integrates both historical and future textual data with numerical time series data. The model is designed to improve forecasting performance by leveraging rich semantic information from textual data, which traditional time series models often lack.

**Strengths:**

1. The paper presents a novel approach to time series forecasting by integrating both historical and future textual data, which is a creative extension of existing multimodal time series models. This work has the potential to significantly impact the field of time series forecasting by demonstrating the value of incorporating textual data.
2. The extensive experiments and ablation studies provide robust evidence of the model's effectiveness and the importance of each component.

**Weaknesses:**

1. Although utilizing predictable future textual information is a commendable effort, I am concerned that in synthetic data and captioned datasets, the inclusion of predictable future textual information may lead to information leakage (where they represent future ground truth), potentially resulting in unfair comparisons with other models and an overestimation of this model's performance.
2. While the paper focuses on time series forecasting, it does not explore the potential of Dual-Forecaster in other multimodal time series analysis tasks (Imputation, Anomaly Detection, etc.). Expanding the scope could provide a more comprehensive evaluation of the model's capabilities.

**Questions:**

Please see the weaknesses.

**Details Of Ethics Concerns:**

The paper does not raise any significant ethical concerns.

---

> ### Author Response · Authors · 2024-11-21
>
> We sincerely thank the comments and suggestions given by the reviewer. We would like to respond to each of the weaknesses you mentioned and hope it will fully address your concerns.
>
> ## **Response to Weakness #1** ##
>
> Thank you for recognizing our utilization of predictive future textual information to assist in time series forecasting. Regarding your concerns about potential information leakage, it is a common response raised by reviewers, please refer to “**Potential Information Leakage**” section in the Common Response. **Our work aims to demonstrate that combining textual data can indeed improve the model’s forecasting performance.** We have validated on the constructed dataset the effectiveness of our model in improving forecasting performance by combining textual data. Our work is a pioneering work in the field of multimodal time series forecasting, and Dual-Forecaster's advanced multimodal comprehension capability and favorable future textual insights-following forecasting ability enable it to be easily extended to practical time series forecasting scenarios.
>
> As for your concerns about potential unfair comparisons, we would like to provide the following explanation: as mentioned at the beginning of the "MAIN RESULTS" section in the paper, regarding the selection of baselines, we choose the four single-modal models, DLinear, FITS, PatchTST, and iTransformer, to demonstrate that integrating textual data can enhance forecasting performance. We acknowledge that without any textual input, our model's predictive performance cannot consistently outperform other single-modal baseline models, **but our model consistently ranks among the top** (the count of the best and the second best results is 4 and 2, respectively, including synthetic dataset and captioned-ETTm1/ETTm2/ETTH1/ETTH2/exchange-rate/stock datasets, where the exchange-rate and stock datasets are the two newly added datasets according to the other reviewer’s suggestions). Additionally, we choose two multimodal models of MM-TFSlib and Time-LLM to highlight the superiority of our model in multimodal information fusion. We acknowledge that these two multimodal models do not utilize future textual data, which may result in unfairness in model performance comparison. **However, we have also conducted comparative experiments using only historical textual data. Experimental results demonstrate that our model consistently outperform these two multimodal models in forecasting performance on the seven datasets mentioned above.**
>
> ## **Response to Weakness #2** ##
>
> **As our model is named, Dual-Forecaster, our work focuses on time series forecasting tasks.** Dual-Forecaster is **a pioneering work** in the field of multimodal time series forecasting, which combines descriptively historical textual information and predictive textual insights to assist in time series forecasting. Through extensive experiments, it has been proven that it can effectively integrate multimodal information, thereby significantly improving the model's forecasting performance. At the same time, it demonstrates advanced multimodal comprehension capability. I fully agree with your suggestion of 'Expanding the scope could provide a more comprehensive evaluation of the model's capabilities'. However, as multimodal time series analysis is a relatively emerging field, to my knowledge, there is still a lack of comprehensive datasets and relevant research. Therefore, there is still a lot of work to be done to extend Dual-Forecaster to other multimodal time series analysis tasks to explore its potential, which is also our future work (see in the "Limitation&Future Work" section). And I believe that the advantage of Dual-Forecaster in multimodal comprehension capability helps it more easily adapt to various time series analysis tasks.

---

> > ### Author Response · Authors · 2024-11-25
> >
> > Dear Reviewer E6mB,
> >
> > We hope that our rebuttal, along with the modifications to the draft, have effectively addressed the weaknesses and questions raised in your review. We are eager to engage in further  questions or concerns regarding our work during the discussion period. If our clarifications and the newly added results align with your expectations, we respectfully request you to consider revising the score.

---

> ### Comment · Reviewer_E6mB · 2024-11-26
>
> I appreciate the authors' feedback. However, the issue of data leakage and the resulting concerns about practical usability remain unresolved.
>
> I understand the authors' claim that the Dual-Forecaster can efficiently capture textual information about future trends, surpassing previous models. However, in real life, it is highly unlikely that we will have access to highly accurate future information. This means that the textual information representing future trends in practical applications is likely to be very inaccurate, leading to a significant discrepancy between the training and testing datasets of the Dual-Forecaster and real-world scenarios. Even if we assume that we can confidently obtain highly accurate textual descriptions of the future, would we then only need a very simple model combined with manual intervention based on these accurate descriptions to make predictions?
>
> In summary, I am concerned that there is a significant gap between the future information used for training and testing and the future textual descriptions that will be available in practical applications, which casts doubt on the actual effectiveness of the Dual-Forecaster.

---

> > ### Author Response · Authors · 2024-11-28
> >
> > Thanks for your responses.
> >
> > We have to clarify that the textual data in all datasets we use is **NOT highly accurate**, including some incorrect information. Since textual data only contains trend information over a specific time period, without amplitude, state changes, and other information, even if we can obtain **relatively accurate** textual descriptions of the future, we cannot simply make predictions based on this textual information. For example, if we know that the sales of a certain product will increase in a certain period of time in the future, **how will the sales of that product increase in the future? Will it first decrease and then increase? Or is it constantly rising? How much has it risen to?** We cannot make accurate predictions based solely on this future trend judgment, but also need to consider the changes in the historical sales data of the product. This is precisely the motivation behind our work: combining textual data to assist time series data for accurate forecasting.
> >
> > Extensive experimental results on datasets containing relatively accurate textual information demonstrate that **Dual-Forecaster is a distinctly effective multimodal time series model**, highlighting the **advanced multimodal comprehension capability and textual instruction-following forecasting ability**.
> >
> > Undoubtedly, validating only on datasets containing relatively accurate textual information cannot determine the actual effectiveness of our model, especially in practical applications where textual information is likely to be inaccurate, which is also a common concern raised by other reviewers.
> >
> > Therefore, to eliminate this concern, we **have supplemented another comparative experiment** with all baselines (including a new baseline of TGForecaster) on **the Weather-captioned dataset** proposed in [1]. As described in [1], this dataset includes weather forecasting reports detailing the climate conditions, which are updated every six hours and daily. The authors employ GPT-4 with designed prompt to summarize these reports into seven brief sentences, which **include somehow noisy or inaccurate information**. For further details, please refer to [1]. The comparative experimental results are shown below.
> >
> > ||Dual-Forecaster||TGForecaster||MM-TSFlib||Time-LLM||DLinear||FITS||PatchTST||iTransformer||
> > |:----|:----|:----|:----|:----|:----|:----|:----|:----|:----|:----|:----|:----|:----|:----|:----|:----|
> > |Metric|MSE|MAE|MSE|MAE|MSE|MAE|MSE|MAE|MSE|MAE|MSE|MAE|MSE|MAE|MSE|MAE|
> > |Weather_captioned|`0.0297`|`0.1174`|0.0600|0.1928|0.0575|0.1792|0.0449|0.1613|0.0677|0.1926|0.0628|0.1849|0.0525|0.1650|0.0558|0.1719|
> >
> > Experimental results show that Dual-Forecaster consistently outperforms other SOTA models, indicating that **our model could still demonstrate excellent forecasting performance in practical applications** where textual information is likely to be inaccurate.
> >
> > [1] TGTSF: Beyond Trend and Periodicity: Guide Time Series Forecasting with Textual Cues.

---

> > > ### Author Response · Authors · 2024-12-02
> > >
> > > Dear Reviewer E6mB,
> > >
> > > We hope the above responses have adequately addressed your concerns. We are eager to engage in further questions or concerns regarding our work during the discussion period. If our clarifications align with your expectations, we respectfully request you to consider revising the score.
> > >
> > > Best regards,
> > > Authors

---

### Official Review · Reviewer_J1Sz · 2024-11-01

**Soundness:** 3
**Presentation:** 3
**Contribution:** 2
**Rating:** 5
**Confidence:** 5

**Summary:**

This paper introduces a time series forecasting model that incorporates both historical and future textual data to improve forecasting accuracy. The proposed model, Dual-Forecaster, uses two main branches: a textual branch for embedding historical and predictive texts and a temporal branch for numerical time series data. It employs three cross-modality alignment techniques—contrastive loss, history-oriented, and future-oriented modality interaction modules—to align and integrate information across these modalities. Experimental results on synthetic and public datasets demonstrate that Dual-Forecaster outperforms baseline models by leveraging multimodal comprehension, thus underscoring the value of textual insights in time series forecasting. The study opens new pathways for integrating diverse data types to enhance predictive performance across various real-world applications.

**Strengths:**

- The paper is well-structured and clearly written, with a logical organization that makes it easy to follow the development of ideas. The authors include illustrative figures, such as the model architecture and case studies, which aid in understanding the Dual-Forecaster model’s interactions and multimodal integration strategies.
- The methodology is clearly presented, and the cross-modality alignment techniques—contrastive loss, history-oriented, and future-oriented interaction modules—are well-integrated within the model. These mechanisms aim to capture complex relationships between text and time series data, contributing to the overall approach.
- The experimental results are strong, including evaluations on synthetic and real-world datasets as well as ablation studies. The results indicate that the proposed components have a positive impact on performance, providing evidence of Dual-Forecaster’s potential effectiveness in multimodal time series forecasting.

**Weaknesses:**

- **Missing Baseline Comparison**: The authors reference the paper "Beyond Trend and Periodicity: Guiding Time Series Forecasting with Textual Cues" by Xu et al. (2024), yet they do not conduct a comparison experiment with this approach. Given the conceptual similarity between the two models, it is crucial to include this as a baseline to demonstrate any distinct advantages of the proposed method.
- **Over-reliance on Synthetic Data and Limited Real-world Testing**: While the paper includes evaluations on both synthetic and public datasets, the reliance on synthetic data limits the applicability of the results. Synthetic setups with artificial textual descriptions do not reflect the complexities and nuances of real-world time series data. The public datasets used (e.g., ETTm1, ETTm2) are also limited in variety and complexity. To strengthen the paper, the authors should include more challenging datasets, such as exchange rates or stock indices, which better represent real-world scenarios. Additionally, the paper lacks evaluation on how the model handles noisy or misleading textual data—an issue highly relevant in real-world, high-stakes contexts.
- **Limited Justification for Dual Textual Modality** The inclusion of both historical (descriptive) and future (predictive) textual information is presented as a core feature, yet the paper provides limited justification for this design choice. It is unclear whether this dual modality substantially enhances forecasting or if similar results could be achieved using only one type of text. A direct ablation study comparing the use of single versus dual text modalities would help clarify this.
- **Error Bounds Missing** The paper does not report error bounds or standard deviations for the results across multiple runs. This omission makes it difficult to assess the consistency and robustness of the reported performance improvements. Including error bounds would provide a more rigorous evaluation of the model's stability.

**Questions:**

- Have you considered simply combining the historical and future texts into a single input rather than using separate modules to handle them? This approach could reduce model complexity. How do you anticipate that combining the texts would affect forecasting performance, and is there evidence that using separate modules significantly enhances results?
- How would your captioning method perform on more complex and volatile datasets, such as exchange rates or stock indices, where trends and patterns are less predictable? Given that these types of data often contain abrupt changes and noise, would your approach to generating textual descriptions still produce reliable or meaningful captions?
- Real-world applications, especially in high-stakes domains like finance, often include noisy or misleading textual data. How does your model handle such cases where the text may not accurately reflect trends in the time series? Have you tested the robustness of Dual-Forecaster with noisy or contradictory textual inputs, and if not, could you discuss any mechanisms that could be added to mitigate potential misdirection from unreliable text?

---

> ### Author Response · Authors · 2024-11-21
>
> We sincerely thank the suggestions given by the reviewer. Below please find the responses to some specific comments and we hope your concerns will be addressed successfully.
>
> ## **Response to Weakness #1** ##
>
> Thank you for the comment. The work in this paper "Beyond Trend and Periodicity: Guiding Time Series Forecasting with Textual Cues" does indeed have some conceptual similarities with our work and is currently one of the few multimodal time series forecasting work. **There are two main reasons why we do not choose the proposed TGForecaster in this work as the baseline:**
> 1. TGForecaster has not yet open-sourced, and although it is easy to reproduce, we are concerned that some details may have been overlooked and cannot objectively reflect its real forecasting performance.
> 2. The textual data used in TGForecaster is the channel description and news corresponding to each future time stamps, while the textual data used in our model Dual-Forecaster is the shape-based descriptions of time series patterns in the entire time period of the look back window and prediction interval, namely the descriptive historical textual information and the predictive future textual insight mentioned in the paper. Therefore, the format of the textual data used by the two models is inconsistent.
>
> In summary, **to achieve fair comparison with each baseline,** we ultimately do not choose TGForecaster as one of the baselines. We will attempt to make a fair comparison with it and include the results in the final version of the paper.
>
> ## **Response to Weakness #2** ##
>
> Thank you for highlighting the potential limitations of relying on synthetic data and the importance of evaluating real-world complexities. We agree that testing on diverse and challenging datasets, as well as considering noisy or misleading textual data, is crucial for assessing the robustness and applicability of our approach. So, we have **added experiments on the exchange-rate and stock dataset.** The experimental results have been added in the revised paper (see Table 2, marked in blue).
>
> As for the misleading textual data issue, please refer to “**Misleading Textual Data**” section in the Common Response.
>
> ## **Response to Weakness #3** ##
>
> Thank you for pointing out this issue. However, in Table 4, we **do indeed provide** ablation experimental results of our model’s forecasting performance using only descriptive historical text or predictive future text as well as using both.
>
> ## **Response to Weakness #4** ##
>
> Thank you for pointing out this question. However, we **do indeed already provide** the Error bars results in Appendix D.

---

> > ### Author Response · Authors · 2024-11-21
> >
> > ## **Response to Question #1** ##
> >
> > Thank you for your valuable suggestion. As mentioned in 3.2.1, since text representation and time series representation are in different high-dimensional spaces, we aligned them before fusing these two modalities. We believe that alignment can bring text representation and time series representation closer together, which is beneficial for converting textual data into a "language" that is easier for time series models to understand and thus be beneficial to obtain high-quality time series representations. This is also the reason why we do not consider simply combining historical and future texts as text inputs for the model, **as only historical time series data can be aligned with the text.** Figure 3 illustrates that our model can achieve accurate alignment between text and time series, and the ablation experimental results in Table 4 demonstrate that the designed Historical Text-Time series Contrastive Loss can indeed improve forecasting performance, indirectly proving the effectiveness of using separate modules.
> >
> > ## **Response to Question #2** ##
> >
> > **We include case studies to showcase our captioning method performing on exchange-rate and stock dataset in the revised paper** (see Figure 6 and Figure 7, marked in blue). These figures demonstrate the stability of our captioning method. Specifically, our captioning method can effectively capture abrupt changes in the data.
> >
> > ## **Response to Question #3** ##
> >
> > Thank you for your valuable suggestion. We acknowledge that we do not test the robustness of our model using noisy or contradictory textual inputs. However, we have to emphasize that we aim to demonstrate that our proposed multimodal information fusion framework does indeed improve the model’s forecast performance, so the textual data utilized in the training phase is accurate. We believe that **training Dual-Forecaster from scratch using text data containing both accurate and incorrect information**, or **fine-tune trained Dual-Forecaster using textual data containing incorrect information**, can mitigate potential misdirection caused by unreliable text. For a detailed explanation, please refer to “**Misleading Textual Data**” section in the Common Response.

---

> > > ### Author Response · Authors · 2024-11-25
> > >
> > > Dear Reviewer J1Sz,
> > >
> > > We hope that our rebuttal, along with the modifications to the draft, have effectively addressed the weaknesses and questions raised in your review. We are eager to engage in further questions or concerns regarding our work during the discussion period. If our clarifications and the newly added results align with your expectations, we respectfully request you to consider revising the score.

---

> > > > ### Comment · Reviewer_J1Sz · 2024-11-25
> > > >
> > > > Thank you for your detailed responses. While I appreciate the effort in addressing the concerns raised, I still find certain explanations and justifications insufficient, as outlined below:
> > > >
> > > > # 1. Comparison with TGTSF
> > > >
> > > > You stated that TGTSF **is not open-sourced, but this is inaccurate**. The TGTSF code is available both in an anonymous repository provided during the review process **(https://anonymous.4open.science/r/TGTSF_review-6E51)** and in the official GitHub repository linked with their arXiv preprint. Given this, your claim about unavailability appears incorrect, and **it raises questions about why TGTSF was excluded as a baseline, especially considering its conceptual similarity to your proposed method**. Including TGTSF would provide a more comprehensive and fair comparison and strengthen the validity of your results. I strongly recommend revisiting this aspect to ensure the comparison is complete and transparent.
> > > >
> > > > # 2. Ablation Studies on Other Datasets
> > > >
> > > > Your ablation study on the model’s ability to forecast without future text description is limited to the ETTm2 dataset. This is insufficient to generalize the robustness and applicability of your model, particularly with the newly added exchange-rate and stock datasets. These datasets represent more complex and realistic forecasting scenarios. Can your model still outperform other baselines in these scenarios when future text descriptions are excluded? A broader set of ablation experiments across all datasets is critical to demonstrate the robustness of your claims.
> > > >
> > > > # 3. Core Contribution and Generalizability
> > > >
> > > > While introducing accurate textual descriptions to improve forecasting is a valuable idea, **it is relatively intuitive that models will perform better with access to additional, accurate information.** The critical challenge lies in demonstrating robustness when such textual data is noisy, misleading, or unavailable. As such, **the scope of your contribution appears limited, especially when your method’s forecasting capability relies heavily on the quality of textual inputs**. Your current experiments do not convincingly address these limitations or showcase your model’s adaptability to more realistic scenarios where textual inputs may be unreliable.
> > > >
> > > > Based on these concerns, I would like to improve my confident score.

---

> > > > > ### Author Response · Authors · 2024-11-28
> > > > >
> > > > > ## **Response to Comment #1** ##
> > > > >
> > > > > Thanks for pointing out this issue. To address your concerns, we **have conducted additional comparative experiments including the new baseline model of TGForecaster** on Weather-captioned dataset (proposed in [1]). Experimental results are shown below.
> > > > >
> > > > > ||Dual-Forecaster||TGForecaster||MM-TSFlib||Time-LLM||DLinear||FITS||PatchTST||iTransformer||
> > > > > |:----|:----|:----|:----|:----|:----|:----|:----|:----|:----|:----|:----|:----|:----|:----|:----|:----|
> > > > > |Metric|MSE|MAE|MSE|MAE|MSE|MAE|MSE|MAE|MSE|MAE|MSE|MAE|MSE|MAE|MSE|MAE|
> > > > > |Weather_captioned|`0.0297`|`0.1174`|0.0600|0.1928|0.0575|0.1792|0.0449|0.1613|0.0677|0.1926|0.0628|0.1849|0.0525|0.1650|0.0558|0.1719|
> > > > >
> > > > > **Experimental results demonstrate that our model consistently outperforms other baseline models, including TGForecaster**. After including TGForecaster as a baseline, our model has been compared with several representative multimodal time series forecasting models (e.g., Time-LLM, MM-TSFlib, and TGForecaster), which **provides a more comprehensive and fair comparison**. Moreover, our model always outperforms them in terms of forecasting performance, demonstrating its effectiveness.
> > > > >
> > > > >
> > > > > ## **Response to Comment #2** ##
> > > > >
> > > > > Thanks for your suggestions. We **do indeed conduct experiments on all datasets for the ablation study on the model’s ability to forecast with/without future textual data**. Below shows the ablation experimental results.
> > > > >
> > > > > ||ETTh1||ETTh2||ETTm1||ETTm2||exchange-rate||stock||
> > > > > |:----|:----|:----|:----|:----|:----|:----|:----|:----|:----|:----|:----|:----|
> > > > > |Metric|MSE|MAE|MSE|MAE|MSE|MAE|MSE|MAE|MSE|MAE|MSE|MAE|
> > > > > |with future textual data|`1.4190`|`0.9134`|`0.8210`|`0.6895`|`1.2126`|`0.7686`|`0.8469`|`0.5756`|`1.8774`|`0.7607`|`0.3239`|`0.3695`|
> > > > > |without future textual data|1.4337|0.9194|0.9429|0.7467|1.3774|0.8222|0.9363|0.6108|2.2011|0.8458|0.4260|0.4358|
> > > > >
> > > > > Experimental results show that our model's forecasting performance has been **further improved on all datasets after inputting future textual data**. We will include these results in the final version of our paper.
> > > > >
> > > > >
> > > > > ## **Response to Comment #3** ##
> > > > >
> > > > > Thanks for your insightful suggestions. To address your concerns, we **have supplemented another comparative experiment** with all baselines (including a new baseline of TGForecaster) on **the Weather-captioned dataset** proposed in [1]. As described in [1], this dataset includes weather forecasting reports detailing the climate conditions, which are updated every six hours and daily. The authors employ GPT-4 with designed prompt to summarize these reports into seven brief sentences, each focus on a specific aspect such as temperature, wind direction, humidity, etc. **These pieces of information may be noisy (irrelevant to the specific channel), misleading (inaccurate), and unavailable**. Take a text description including "Without fine grained details, temperature trends are not specific" in the Weather-captioned dataset as an example. It indicates that current information about temperature trends is unavailable. Regarding the experimental results, please refer to our response to the comment of **“1. Comparison with TGTSF”** mentioned above.
> > > > >
> > > > > Experimental results show that our model consistently outperforms other SOTA models (including TGForecaster), indicating that our model can still adapt well in real-world forecasting scenarios where textual inputs may be unreliable and demonstrate excellent forecasting capability.
> > > > >
> > > > > [1] TGTSF: Beyond Trend and Periodicity: Guide Time Series Forecasting with Textual Cues.

---

> > > > > > ### Author Response · Authors · 2024-12-02
> > > > > >
> > > > > > Dear Reviewer J1Sz,
> > > > > >
> > > > > > We hope the above responses have adequately addressed your concerns. We are eager to engage in further questions or concerns regarding our work during the discussion period. If our clarifications align with your expectations, we respectfully request you to consider revising the score.
> > > > > >
> > > > > > Best regards,
> > > > > > Authors

---

### Official Review · Reviewer_uK6h · 2024-11-04

**Soundness:** 2
**Presentation:** 3
**Contribution:** 2
**Rating:** 3
**Confidence:** 4

**Summary:**

The paper introduces a novel text-guided time series forecasting model, named Dual-Forecaster, which leverages both historical and predictive textual data to enhance forecasting accuracy. This model has been evaluated using synthetic and real-world datasets, demonstrating superior performance.

**Strengths:**

- The paper is well-organized and easy to follow.

- The model architecture is well explained and seems to be solid.

- The ablation study, particularly the alignment heatmap between time series patterns and textual descriptions, provides in-depth understanding on how the model actually works.

**Weaknesses:**

- The task definition may need further discussion. The statement right now is clear. However, the practicality of the task definition could be better justified, specifically regarding the availability of caption data in real-world applications.

- There may be potential information leakage in this setting/dataset used. According to the TGTSF [1] cited in related work, to avoid directly using the information in the future time series, it is advisable to use the text information from external sources that is related to the system we are analyzing.

- The captioning process of public dataset needs further discussion. Directly asking the GPT to describe the time series may not be a promising method.
    - The GPT's ability of processing numerical values still remains under discussion. Asking it to generate description of time series which is also a sequence of float values, may suffer from severe hallucination problem.
    - As mentioned above, since the textual descriptions are derived directly from the time series data, there is a risk of information leakage, particularly if these descriptions are used for future predictions.
    - It seems that the captioned textual information also contains numerical values. It is still questionable that if the information in these values still remains after the text embedding model, i.e. RoBERTa in this paper.

- According to the ablation study, this work seems to fall into the category of aligning time series pattern and textual description. Maybe it is a better idea to apply this to classification and anomaly detection task, which is not very sensitive to the information leakage problem.

[1] Zhijian Xu, Yuxuan Bian, Jianyuan Zhong, Xiangyu Wen, and Qiang Xu. Beyond trend and periodicity: Guiding time series forecasting with textual cues.

Minor:

- Some figures are hard to read, e.g. Figure 3 & 5.

**Questions:**

- How are the future-oriented predictive text generated? It seems that authors use the known time series for caption to generate the description for trend and periodicity. This is reasonable for the look-back window part of each training sample, but using the caption for forecasting horizon may lead to severe information leakage.

- See Weakness.

---

> ### Author Response · Authors · 2024-11-21
>
> We thank reviewer's insightful suggestions. Hope the following response can address your concerns.
>
> ## **Response to Weakness #1** ##
>
> Thank you for your suggestion. In real-world time series forecasting scenarios, in addition to historical time series data, **there also generally exists context information related to the scenarios (usually in text form)**, e.g., news, product promotion plans. Although this information cannot be directly correlated with future trends, **it can always be converted into the judgment of future trends through specific methods**, such as prompting large language model to summarize or making judgments based on experts’ experience. Our model can directly utilize the generated judgements of future trends, which can provide forecasts that are more in line with actual business expectations through its advanced multimodal comprehension capability and favorable future text insights-following forecasting ability. Extensive experimental results have proven it. Additionally, our model can also be easily adapted to directly combine context information for forecasting, which is our future work.
>
> ## **Response to Weakness #2 & Weakness #4 & Question #1** ##
>
> Regarding your concern about information leakage, we provide a unified reply. Please refer to the "**Potential Information Leakage**" section of the Common Response.
>
> ## **Response to Weakness #3** ##
>
> Thank you for your insightful question. First, I would like to clarify that for the synthetic dataset, **we only used GPT to paraphrase annotations that were initially generated based on rules**, aiming to enhance linguistic diversity. We do not utilize GPT's annotation capabilities for time series data. For the real-world dataset, we do not employ GPT for data annotation either; instead, we used a sequence segmentation algorithm combined with statistical computations to derive rule-based textual descriptions.
>
> Your question regarding the preservation of numerical information in text embedding models is highly significant. Through visualization of specific forecasting cases (Figures 8 to Figure 14), we observe that the forecasts exhibit the ability to follow instructions concerning numerical information. Additionally, the attention maps presented in Figure 15 indicate that numerical information is effectively utilized. Overall, while pre-trained language models such as RoBERTa are not specifically designed for numerical processing, they are capable of implicitly capturing relationships involving numerical values through contextual learning to a certain extent.
>
> ## **Response to Minor** ##
>
> We have added more details to Figures 3 and Figure 5 and provided additional information about our real-world time series captioning method in the Appendix B.2.2 to clarify the captioning process (marked in red).

---

> > ### Author Response · Authors · 2024-11-25
> >
> > Dear Reviewer uK6h,
> >
> > We hope that our rebuttal, along with the modifications to the draft, have effectively addressed the weaknesses and questions raised in your review. We are eager to engage in further questions or concerns regarding our work during the discussion period. If our clarifications and the newly added results align with your expectations, we respectfully request you to consider revising the score.

---

> > > ### Comment · Reviewer_uK6h · 2024-11-25
> > >
> > > We appreciate the authors’ efforts to address our questions. While some of our concerns have been adequately clarified, the issue of information leakage remains unresolved.
> > >
> > > We commend the authors for introducing the concept of “information insufficiency” in the introduction, which is insightful. However, our primary concern continues to center on the generation of “predictive text.”
> > >
> > > In the common response, the authors state, “we construct future textual data based on the time series in the prediction interval, which contains accurate future trend information.” However, within the scope of this work, the predictive texts are generated by asking an LLM to describe the time series. This implies that one must already have or know the future time series to generate these descriptions. This raises a critical question: if the exact future patterns are already known, what is the purpose of performing forecasting?
> > >
> > > The authors also argue that “any other form of textual data (such as news, weather forecasting reports, etc.) can be summarized into similar textual descriptions through specific methods, which can always be obtained before forecasting.” However, as noted in the response to Weakness 1, summarizing such general textual data into shape-based descriptive text requires significant domain knowledge and manual effort. Alternatively, relying on LLMs would necessitate embedding this knowledge into the models, which presents another open challenge. Furthermore, even with these efforts, achieving the precise shape-based descriptive text required by Dual-Forecaster is not guaranteed. The paper also lacks experimental evidence to substantiate this claim.
> > >
> > > This leads to another concern: according to Point 3 in the common response, the authors aim to enable the model to learn the correct time series shape in response to external events. However, the model presented in the paper focuses solely on shape description and time series pattern alignment. The transformation of external events into shape descriptions is neither included in the model nor demonstrated in the paper. This further supports our assertion in Weakness 4 that the model is designed for pattern-description alignment, and claims of directly reacting to external events may be overstated.
> > >
> > > If the model’s goal is indeed to achieve direct causal alignment with external events, this task would closely align with the cited works TGTSF and TGForecaster. These works should therefore be included as baselines for comparison, as they have already demonstrated feasible approaches to this objective. However, in the response to Reviewer J1Sz, the authors argue that TGForecaster directly uses channel descriptions and news, while Dual-Forecaster employs shape-based descriptions, making them incomparable. This further suggests that Dual-Forecaster’s alignment with the stated motivation may be questionable.
> > >
> > > Finally, as highlighted in Weakness 1, while the method shows promise, the task definition needs further refinement to better align with realistic scenarios. Given these unresolved concerns, I will maintain my original score.

---

> > > > ### Author Response · Authors · 2024-11-28
> > > >
> > > > Thanks for pointing out these issues.
> > > >
> > > > We have to emphasize that the motivation behind our work is to **combine textual data to assist time series data for accurate forecasting, including descriptively historical texts and predictive texts**.
> > > >
> > > > When only incorporating descriptively historical texts, the comparative experimental results of our model with other baseline models are shown below.
> > > >
> > > > ||Dual-Forecaster||MM-TSFlib||Time-LLM||DLinear||FITS||PatchTST||iTransformer||
> > > > |:----|:----|:----|:----|:----|:----|:----|:----|:----|:----|:----|:----|:----|:----|:----|
> > > > |Metric|MSE|MAE|MSE|MAE|MSE|MAE|MSE|MAE|MSE|MAE|MSE|MAE|MSE|MAE|
> > > > |ETTh1|`1.4337`|`0.9194`|1.4967|0.9347|1.5919|0.9914|1.4999|0.9505|1.6004|0.9952|1.6009|0.9603|1.5128|0.9438|
> > > > |ETTh2|`0.9429`|`0.7467`|0.9616|0.7644|1.0586|0.8083|0.9951|0.7847|1.2858|0.8875|1.0349|0.7879|0.9803|0.7703|
> > > > |ETTm1|1.3500|0.8103|1.3620|0.8426|1.4457|0.8730|1.5601|0.9198|2.2858|1.1810|1.4544|0.8619|`1.3393`|`0.8299`|
> > > > |ETTm2|`0.9363`|`0.6108`|1.0325|0.6691|1.1199|0.7054|1.1663|0.7332|1.7418|0.9709|0.9419|0.6280|1.0210|0.6557|
> > > > |exchange-rate|`2.2011`|`0.8458`|2.6365|1.0061|3.0564|1.1111|3.1668|1.1146|4.4656|1.4831|2.2656|1.0016|2.6426|0.9977|
> > > > |stock|`0.4260`|`0.4358`|0.5256|0.5038|0.4900|0.4866|0.7554|0.6330|1.1194|0.8057|0.4936|0.4755|0.5135|0.4926|
> > > >
> > > > Experimental results show that our model is superior or comparable to other baseline models, indicating that our model can **efficiently utilize additional textual information** through the text and time series alignment module (as evidenced by the text and time series alignment result provided in Figure 3), thereby **enhancing the model’s forecasting ability**.
> > > >
> > > > When further combining predictive texts, our model’s forecasting performance is further improved. The experimental results are shown in Table 2. Experimental results show that our model consistently outperforms other baseline models, demonstrating **impressive textual instructions-following forecasting capability**.
> > > >
> > > > We have to clarify that the textual data in all datasets we use is **NOT exact accurate**, including some incorrect information. Since textual data **only contains trend information over a specific time period, without amplitude, state changes**, and other information, even if we can obtain relatively accurate textual descriptions of the future, we cannot simply make predictions based on this textual information. For example, if we know that the sales of a certain product will increase in a certain period of time in the future, how will the sales of that product increase in the future? Will it first decrease and then increase? Or is it constantly rising? How much has it risen to? We cannot make accurate predictions based solely on this future trend judgment, but also need to consider the changes in the historical sales data of the product. **This is precisely where the value of our work lies:** providing an effective multimodal information fusion framework that enables the use of textual data to assist time series data for more accurate forecasting.

---

> > > > > ### Author Response · Authors · 2024-11-28
> > > > >
> > > > > **We believe that valuable predictive texts can enhance forecasting**. Experimental results also demonstrate that our model can effectively improve forecasting performance by utilizing relatively accurate shape-based descriptive text. **Although this form of relatively accurate descriptive text is difficult to obtain in the real world, it does not mean that there is none**. For example, in the retail industry, experts will obtain relatively accurate trend information for specific products in the future based on their experience, which often helps to provide forecasts that are more in line with business expectations. Undoubtedly, validating only on datasets containing relatively accurate predictive texts cannot determine the actual effectiveness of our model, especially in practical applications where accurate predictive information is difficult to obtain.
> > > > >
> > > > > Therefore, to address your concerns, we **have supplemented another comparative experiment** with all baselines (including a new baseline of TGForecaster) on **the Weather-captioned dataset** proposed in [1]. As described in [1], this dataset includes weather forecasting reports detailing the climate conditions, which are updated every six hours and daily. The authors employ GPT-4 with designed prompt to summarize these reports into seven brief sentences, which **include somehow noisy or inaccurate information**. The comparative experimental results are shown below.
> > > > >
> > > > > ||Dual-Forecaster||TGForecaster||MM-TSFlib||Time-LLM||DLinear||FITS||PatchTST||iTransformer||
> > > > > |:----|:----|:----|:----|:----|:----|:----|:----|:----|:----|:----|:----|:----|:----|:----|:----|:----|
> > > > > |Metric|MSE|MAE|MSE|MAE|MSE|MAE|MSE|MAE|MSE|MAE|MSE|MAE|MSE|MAE|MSE|MAE|
> > > > > |Weather_captioned|`0.0297`|`0.1174`|0.0600|0.1928|0.0575|0.1792|0.0449|0.1613|0.0677|0.1926|0.0628|0.1849|0.0525|0.1650|0.0558|0.1719|
> > > > >
> > > > > Experimental results show that Dual-Forecaster consistently outperforms other SOTA models, indicating **our model’s validity in practical applications where textual information is likely to be inaccurate**.
> > > > >
> > > > > [1] TGTSF: Beyond Trend and Periodicity: Guide Time Series Forecasting with Textual Cues.

---

> > > > > > ### Author Response · Authors · 2024-12-02
> > > > > >
> > > > > > Dear Reviewer uK6h,
> > > > > >
> > > > > > We hope the above responses have adequately addressed your concerns. We are eager to engage in further questions or concerns regarding our work during the discussion period. If our clarifications align with your expectations, we respectfully request you to consider revising the score.
> > > > > >
> > > > > > Best regards,
> > > > > > Authors

---

> > > > > > > ### Comment · Reviewer_uK6h · 2024-12-02
> > > > > > >
> > > > > > > Dear Authors,
> > > > > > >
> > > > > > > I appreciate the effort author made to further address our concern. However, the author still not directly solve my information leakage concern of using *descriptive* text.
> > > > > > >
> > > > > > > I strongly agree with the idea of incorporating the text information can enhance the forecasting performance and I really appreciate the superior performance Dual-forecaster delivers with the descriptive text. It effectively demonstrate its capability of aligning the time series patterns with text description.
> > > > > > >
> > > > > > > My concerns still lies in the *descriptive* part. As I mentioned, to generate the description of the future trend, someone must get access to the future time series which will cause information leakage. But if someone can make trend prediction out of some foreseeable events, then why not use the original TGTSF to let the model learn the causal relationship. This is a dilemma I think authors have to address.
> > > > > > >
> > > > > > > Especially, the added experiment show that the Dual-Forecaster can achieve extraordinary performance on the TGTSF dataset. This show that the Dual-Forecaster is a very powerful text-guided time series forecasting model. I think this shows that even without the descriptive text that may lead to leakage problem, your model is still powerful. So maybe just selling it as a follow-up TGTSF model can avoid the above dilemma.
> > > > > > >
> > > > > > > About the added experiment. It seems that author did not provide any experiment setting. And the performance of TGforecaster is way better than the ones reported in TGTSF paper. Is it because of the different setting or the author actually still use the trend description to guide these two models that leaks more accurate information?
> > > > > > >
> > > > > > > Overall, in the discussion, **authors have successfully convince me that their model to be a powerful text-guided time series forecasting model and I really appreciate some of their insights**. But the explanation of the major novelty of "using descriptive text" is still pale, Nevertheless it introduce the information leakage dilemma that many reviewers also raise concerns about. Especially, some of the experimental results (if conducted correctly) further evident that it can be a powerful and effective TGTSF model even without such descriptions.
> > > > > > >
> > > > > > > Thus, even though I like the method authors proposed, some of the major claims still have contradiction and beyond revision. I regretfully have to keep my score. Believe me, **this is not a deliberate toxic decision** even though the academic environment now is not friendly.  And I sincerely believe this will be a great work with some appropriate task setting.
> > > > > > >
> > > > > > > Sincerely,
> > > > > > > Reviewer
> > > > > > >
> > > > > > > P.S. I decide not to use the GPTs to help me write this response to show my sincereness.

---

> > > > > > > > ### Author Response · Authors · 2024-12-03
> > > > > > > >
> > > > > > > > Dear Reviewer uK6h,
> > > > > > > >
> > > > > > > > Thanks very much for your sincere reply and your affirmation of our work.
> > > > > > > >
> > > > > > > > Firstly, we would like to **provide the experimental settings for the newly added experiment**. We set the look back window length as 288 (consistent with that reported in TGTSF paper), and the predicted horizon is fix as 36 (according to the description of Weather-captioned dataset, at the current time, we can only know the weather conditions for the next 6 hours in advance, and 6 hours correspond to 36 10-minute intervals. We believe that setting the predicted horizon to 96 or greater may have information leakage issues, so we choose to set the predicted horizon to 36. This setting is inconsistent with that reported in TGTSF paper). **Regarding the textual data, we directly use the original texts from the Weather-captioned dataset, rather than trend descriptions**. We will add the setup instructions for this experiment to the final version of the paper.
> > > > > > > >
> > > > > > > > Secondly, we would like to emphasize once again that **the main task of our work is how to efficiently utilize textual data to enhance the model’s forecasting performance**. To verify the actual effectiveness of our model, we meticulously design verification experiments step by step. **The idea of our experimental design is as follows**:
> > > > > > > >
> > > > > > > > 1. **“From 0 to 1”: that is, to verify whether our model can truly utilize textual data and whether utilizing textual data can truly enhance forecasting performance**. In this step, we use descriptively historical text data for validation. The result of historical text-time series alignment shown in Figure 3 demonstrates that our model can indeed effectively understand and utilize the semantic information contained in textual data. Additionally, the comparative experimental results with single-modal baseline models demonstrate that combining textual data can indeed improve forecasting performance.
> > > > > > > >
> > > > > > > > 2. **“From 1 to better”: that is to verify the advancement of our model in combining textual data to assist time series data for forecasting**. In this step, we also use descriptively historical text for verification. Compared with the two multimodal models of Time-LLM and MM-TFSlib, our model has better forecasting performance, indicating its superiority in effectively incorporating textual data for more accurate forecasting. Thus, our model has been strongly validated in combining descriptively textual data to assist in more accurate time series forecasting. **Whether it is in the concept itself (combining textual data for time series forecasting) or in the meticulously designed model framework, it is one of the highlights presented in this article. From this perspective, we believe that the contributions of our work cannot be underestimated**.

---

> > > > > > > > > ### Author Response · Authors · 2024-12-03
> > > > > > > > >
> > > > > > > > > 3. **“From better to more reliable”: that is, to verify whether our model can follow future texts and provide reliable forecasts that meet expectations, namely the ability of predictive textual insights-following forecasting, which is essential in practical applications**. In this step, we further utilize predictive texts for validation based on the above experiment. To verify the ability of our model’s capability of predictive textual insights-following forecasting more intuitively, we design simple textual data that matches future time series data. The forecasting visualization result shown in Figure 2 demonstrates that our model can follow future textual instructions and providing reliable forecasts which reflect state transitions in accordance with textual descriptions. Additionally, from the attention map shown in Figure 15, it demonstrates that our model can indeed capture key information in future texts to guide reliable forecasting, thus proving that our model has the ability of predictive textual insights-following forecasting. **We have to clarify that in this article, the original intention of constructing and utilizing this type of predictive texts is more to verify whether the model can truly achieve predictive textual instructions-following forecasting. If it can be truly achieved, the improvement of forecasting performance is a natural thing. We do indeed verify this point**. Regarding your concerns of “if someone can make trend predictions out of some foreseeable events, Then why not use the original TGTSF to let the model learn the causal relationship?”, we totally agree with your viewpoint, but **we believe that utilizing the simple textual data we construct may be a more intuitive way to verify whether the model has the ability to follow predictive text instructions for forecasting. Moreover, in terms of verifying whether our model can follow predictive text instructions for forecasting, in addition to judging based on MSE or MAE metrics, simple textual descriptions can also serve as a label aid for more direct judgments**.
> > > > > > > > >
> > > > > > > > > 4. **“From reliability to more practical”**: We agree with your and other reviewers' views that our approach of constructing predictive textual data lacks practical operability. Therefore, to address your concerns and polish our work, we **have conducted experiments on the Weather-captioned dataset**. Experimental results indicate that our model also demonstrates excellent forecasting ability for practical application scenarios, **proving the actual effectiveness of our model**.
> > > > > > > > >
> > > > > > > > > The above is a detailed descriptions of our idea of experimental design. On the one hand, we hope to adequately address your concerns. On the other hand, we also want to emphasize that **the original intention of constructing and utilizing this type of predictive texts is mainly to intuitively verify the model's ability to follow future textual instructions, which is esstial in practical applications**. We will make appropriate adjustments to the content of the paper to avoid unnecessary misunderstandings and reflect them in the final version of the paper.
> > > > > > > > >
> > > > > > > > > Finally, thank you again for your valuable feedback and sincerity!
> > > > > > > > >
> > > > > > > > > Best Regards,
> > > > > > > > >
> > > > > > > > > The Authors

---

### Author Response · Authors · 2024-11-21
**Common Response #1: Misleading Textual Data**

**We have to emphasize that our work aims to illustrate that integrating textual data does indeed improve forecast performance.** Therefore, we focus on designing a multimodal information fusion framework with advanced multimodal comprehension capability. To verify the effectiveness of the framework, it is necessary to eliminate the influence of data caused by inaccurate textual information. Therefore, the data we use contains accurate textual information. Experimental results demonstrate the superiority of our model in multimodal information fusion, with advanced multimodal comprehension capability. Moreover, the model enables textual insights-following forecasting, thereby providing more reliable forecasts.

As for the common response of Misleading Textual Data raised by reviews, I will answer from the following two aspects:
1. After training Dual-Forecaster using data containing accurate textual information, during the inference phase, if the textual data contains incorrect information, the Dual-Forecaster will tend to provide forecasts that match the textual information due to its ability of textual insights-following forecasting, which may deviate from the true value.
2. If we **train our model using textual data that contains both accurate and incorrect information**, the model will adaptively learn the forecasting mode under both accurate and incorrect textual data inputs during the training process. During the inference stage, if the input textual data contains incorrect information, the model can still provide accurate forecasts. Additionally, using textual data containing inaccurate information to **fine-tune Dual-Forecaster** can also enable the model to provide accurate forecasts even when inputting textual data containing inaccurate information.

In summary, our current work falls under the first point mentioned above. When inputting textual data containing inaccurate or misleading information, the model's forecasts may deviate from the true values. **The second point mentioned is our future work.** We believe that Dual-Forecaster can be easily extended to time series forecasting scenarios involving textual data input with inaccurate information and provide accurate forecasts.

---

### Author Response · Authors · 2024-11-21
**Common Response #2: Potential Information Leakage**

Regarding the common concern of potential information leakage raised by reviewers, we would like to make the following clarifications:
1. We acknowledge that using information within the prediction interval to assist time series data for forecasting would result in information leakage in traditional time series forecasting framework.
2. However, as we mentioned in the INTRODUCTION section, the current traditional time series forecasting models that rely solely on time series data are **constrained by the limitation of insufficient information**, which has reached a bottleneck in improving the forecasting performance. Therefore, it is crucial to introduce supplementary data to assist time series data for accurate forecasting.
3. The reason why these supplementary data can further improve the model’s forecasting capability is attributed to its ability to **provide more information that is not included in time series data**, such as relatively accurate predictions of future trends. This information is beneficial for accurately grasping future trends and providing more reasonable forecasts, which are also **more in line with the needs of practical time series forecasting scenarios**. However, traditional time series forecasting models are unable to provide accurate forecasts due to the lack of this information when facing distribution changes caused by external events.
4. To simulate actual time series forecasting scenarios, we construct future textual data based on the time series in prediction interval, which contains accurate future trend information. It should be emphasized that the future textual data we constructed does not contain any information reflecting the amplitude of the predicted values (see Figures 6 and Figure 7). Moreover, the future textual data we use is a shape-base textual description of time series trend, which is a relatively universal textual description. Any other form of textual data (such as news, weather forecasting reports, etc.) can be summarized into similar textual descriptions through specific methods, which can always be obtained before forecasting. **Therefore, we believe that there is no information leakage from the perspective of the requirements of real-world time series forecasting scenarios.**

---

### Note · Authors · 2025-01-23

I have read and agree with the venue's withdrawal policy on behalf of myself and my co-authors.